# GEO-R1: UNLOCKING VLM GEOSPATIAL REASONING WITH CROSS-VIEW REINFORCEMENT LEARNING

## ABSTRACT

We introduce Geo-R1, a reasoning-centric post-training framework that unlocks geospatial reasoning in vision-language models by combining thinking scaffolding and elevating. In the scaffolding stage, Geo-R1 instills a "geospatial thinking paradigm" via supervised fine-tuning on synthetic chain-of-thought exemplars, enabling models to connect visual cues with geographic priors without costly human reasoning annotations. In the elevating stage, it uses GRPO-based reinforcement learning on a weakly-supervised cross-view pairing proxy. This design supplies a verifiable and scalable reward signal: teaching models to capture and reconcile features across modalities, and harnessing reasoning for accurate prediction. Geo-R1 extends geospatial modeling from domain pretraining / supervised finetuning to reasoning-first post-training, and achieves state-of-the-art performance across various geospatial reasoning benchmarks. Our code is available at `https://anonymous.4open.science/r/Geo-R1-ICLR`.

## 1 INTRODUCTION

Geospatial reasoning is fundamental to a wide range of scientific and societal applications, spanning disaster response, search and rescue, urban planning, environmental monitoring, and sociocultural study. Unlike common vision-language reasoning (Li et al., 2024) centering around object recognition, captioning and general question-answering, geospatial reasoning spans many modalities (e.g., aerial imagery, streetview photos, location metadata, place information, etc.), and varied tasks (e.g., geographical, environmental, sociocultural, etc.) as shown in Fig. 1. This blend of multimodal evidence and knowledge-intensive tasking makes general reasoning both crucial for geospatial understanding, and also uniquely challenging.

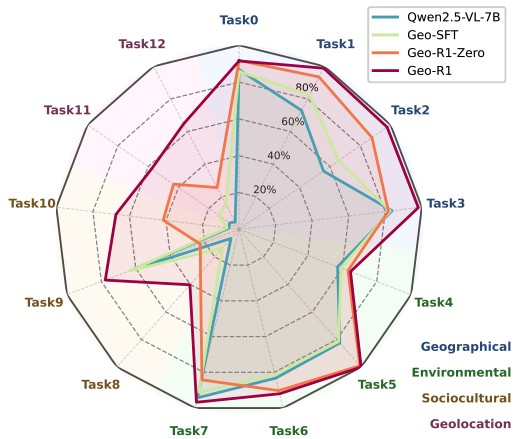

Figure 1: Geo-R1 significantly outperforms baseline Bai et al. (2025) across 13 verifiable geo-reasoning tasks on the GeoChain benchmark (Yerramilli et al., 2025) in the zero-shot setting. See Table 9 for detailed description of these tasks.

Prior geospatial VLMs primarily adopt *supervised fine-tuning (SFT)*. While effective in natural domains, SFT is poorly suited in geospatial settings. Geospatial raw data can be plentiful, but supervisions are sparse, usually limited to coordinate metadata without descriptive content. As a result, SFT-heavy geospatial VLMs consistently display three key failure modes: (1) brittle in-domain stability, (2) limited out-of-distribution generalization, and (3) catastrophic forgetting. For example, a common label source in the geospatial field is satellite object detection datasets, such as DOTA (Xia et al., 2018). But their narrow class coverage produces severe data imbalance. Consequently, many SFT models trained on these data sources (Kuckreja et al., 2024; Zhang et al., 2024a; Muhtar et al., 2024; Pang et al., 2025) generalize poorly, failing on non-detection tasks, degrading under resolution shifts, and losing basic capability in natural domains. Attempts to expand data diversity, such as TeoChat (Irvin et al., 2024), which leverages temporally varied datasets, still suffer performance degradation under small domain drifts.

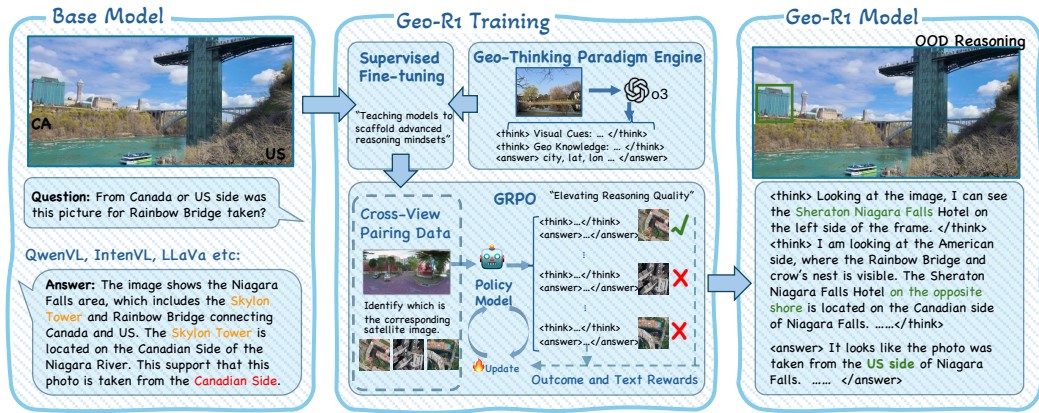

Figure 2: Geo-R1 overview. Geo-R1 provide a framework for building geospatial reasoning.

Recent advances in *reinforcement learning with verifiable rewards (RLVR)* for large reasoning models offer a promising alternative. Models such as DeepSeek-R1 (Guo et al., 2025) demonstrate that post-training with rule-based RL can substantially improve reasoning capabilities, enabling inference-time scaling without dense supervision of intermediate thinking process.

The success of RLVR in math and code domains motivates us to explore similar strategies in geospatial reasoning to relieve the dependence on dense annotations. In our investigation, we encountered several challenges: (1) *Geo-tasking heterogeneity*. Geospatial reasoning spans diverse question families which require distinct input-output patterns and reasoning needs; (2) *Weakly-supervised rewards*. Geospatial supervision is often limited to coarse metadata, offering far less guidance than richly annotated visual language datasets; Both factors lead to challenges for (3) *Verifiability at scale*, e.g., lacking definitive ground-truth answers, or involving subjective judgments in contrast to mathematics. These issues highlight the difficulty of scaling geospatial RLVR, collectively resulting in the absence of a unified framework supporting geospatial reasoning.

We introduce Geo-R1, the first reasoning-centric post-training framework for open-ended geospatial reasoning that addresses aforementioned challenges. Geo-R1 represents a strategic shift away from conventional domain-specific SFT towards a "reasoning-first" post-training, which harmonizes the complementary strengths of SFT for paradigm-level learning and RLVR for outcome-oriented learning. As illustrated in Fig. 2, in the SFT stage, a small amount of carefully constructed synthetic geospatial chain-of-thoughts (CoT) data is used for "geospatial thinking paradigm" training, scaffolding the base model with structural geo-reasoning capabilities, while without inducing catastrophic forgetting. In the RLVR stage, Geo-R1 designs a novel verifiable reward framework that needs only location metadata to elevate model reasoning quality toward accurate outcomes.

Central to the reward framework is a cross-view pairing task: given a streetview or panoramic image, the model must identify the corresponding satellite image from multiple visually similar candidates. This reward provides multiple benefits. (1) *One unified reward for heterogeneous tasks*: rewarded by paring accuracy, we found RLVR greatly enhance VLM's several general-purpose reasoning capabilities, including tiny visual cue extraction (like OCR), cross-view reasoning (like object correspondence), cross-modality information synthesis (like recalling specific place information from pretrained memorization), etc. These capabilities are fundamental to various geospatial reasoning tasks. (2) *Weakly-supervised rewards to motivate strong reasoning behaviors*: unlike generic landmark geolocation, our task uses random streetview images globally and confusing candidate satellite images from same cities, therefore requires substantially more complex reasoning to synthesize every possible visual cues to answer correctly. (3) *Verifiability at scale*: The location metadata is generally available for most image sources across different modalities to achieve RLVR scalability.

We conduct extensive experiments with base `Qwen2.5-VL-7B` (Bai et al., 2025) and our post-trained `Geo-R1` model. Fig. 1 highlighted our zero-shot Geochain benchmark performance improvement. `Geo-R1` demonstrates significant across-the-board improvements from in-distribution to out-of-distribution geospatial reasoning generalization. Meanwhile, strict general-purpose VLM benchmarks are conducted to evaluate catastrophic forgetting. Results show our Geo-R1 post-training effectively preserves the original VLM capabilities (e.g. math-reasoning, OCR, VQA, etc).

## 2 RELATED WORKS

**Geospatial Foundation Models.** Recent geospatial foundation model training paradigms span from general-purpose visual pretraining, e.g., masked auto-encoding (Cong et al., 2022; Szwarcman et al., 2024; Reed et al., 2023), to contrastive learning (Zhang et al., 2024b; Li et al., 2023; Liu et al., 2024b) and remote sensing VLMs (Kuckreja et al., 2024; Muhtar et al., 2024; Pang et al., 2025). While these models excel at specific tasks like representation learning, detection, retrieval, geospatial VQA, most cannot conduct reasoning (e.g., decomposing a task into bearings, distances, landmarks and synthesize information) nor do they refine thinking process to make final decisions.

**Vision Language Model with Chain-of-Thoughts.** Early reasoning VLMs built on chain-of-thoughts and multi-step reasoning for predictions rather than single-step. They augment model performance with chain-of-thought style traces, self-consistency sampling, and programmatic verifiers (Wei et al., 2022; Zhang et al., 2023; Wang et al., 2023). Recent models demonstrate that structured intermediate text, such as plans, sketches, or symbolic arguments improves robustness and out-of-distribution generalization (Cobbe et al., 2021; Schick et al., 2023).

**Inference Time Scaling with Reinforcement Learning.** Inference-time scaling improves reasoning by allocating more test-time compute via RL-based post-training. Outcome- and process-based rewards, human/AI feedback (RLHF/RLAIF), and verifier-guided RL have been shown to produce longer, more structured chains with higher factuality (Christiano et al., 2017; Bai et al., 2022; Kumar et al., 2025). Recent "R1-style" systems train policies (LLMs/VLMs) to generate and self-verify solutions with group-relative or advantage-normalized objectives to stabilize long-horizon updates (Guo et al., 2025; Shao et al., 2024).

Most aforementioned works target natural scene VQA, chart/table reasoning, or math/logic. In this work, we position geospatial reasoning as a primary goal: the model must reason across views, decompose the task into geo-primitives, and justify decisions with intermediate thoughts. To scale up RL in geospatial settings, verifiers should be programmatic and precise, enabling dense, low-latency rewards without human annotation. Our approach integrates inference-time scaling (self-consistency and verifier-guided search) with RL that learns to self-explore intermediate states that improve geo-metrics, closing the loop between perception, reasoning, and measurable correctness.

## 3 GEO-R1: SCAFFOLDING & ELEVATING GEOSPATIAL REASONING

As shown in Fig. 2, Geo-R1 is conceptualized as a two-stage methodology engineered to unlock sophisticated geospatial reasoning capabilities in pre-trained VLMs. The approach harmonizes two philosophically distinct stages: (1) an **scaffolding** stage, which leverages small-scale SFT to instill a structured "geo-thinking" paradigm, and (2) an **elevating** stage, which employs larger-scale RLVR to refine the model's reasoning for factual correctness and conciseness through a verifiable proxy task. This two-stage design directly addresses several critical challenges in the field: the absence of innate geospatial reasoning capabilities in general-domain VLMs, the scarcity of expert-annotated reasoning datasets, and the difficulty in formulating a direct, verifiable reward signal for complex, open-ended geospatial reasoning tasks. We discuss the details in Sec. 4 and Sec. 5 seperately.

## 4 GEO-R1 STAGE#1: GEOSPATIAL THINKING SCAFFOLDING WITH SFT

The first stage of the Geo-R1 framework is dedicated to geospatial thinking **scaffolding**, based on the principle that a model must first learn structured, domain-appropriate reasoning skillsets before it can be effectively optimized for accuracy at scale. This SFT phase is carefully designed to scaffold coherent geospatial reasoning, providing the model with an initial foundation of reasoning.

### 4.1 GEOSPATIAL THINKING SCAFFOLDING

Towards the goal of building a cognitive scaffold for geospatial reasoning, we decide a principle that teaching *domain-generic* reasoning paradigms is more valuable than supervising *question-specific* reasoning and answers, the latter of which can be too diverse for both model learning and data collection at scale. Accordingly, we do not choose to synthesize diverse CoTs for all tasks used in

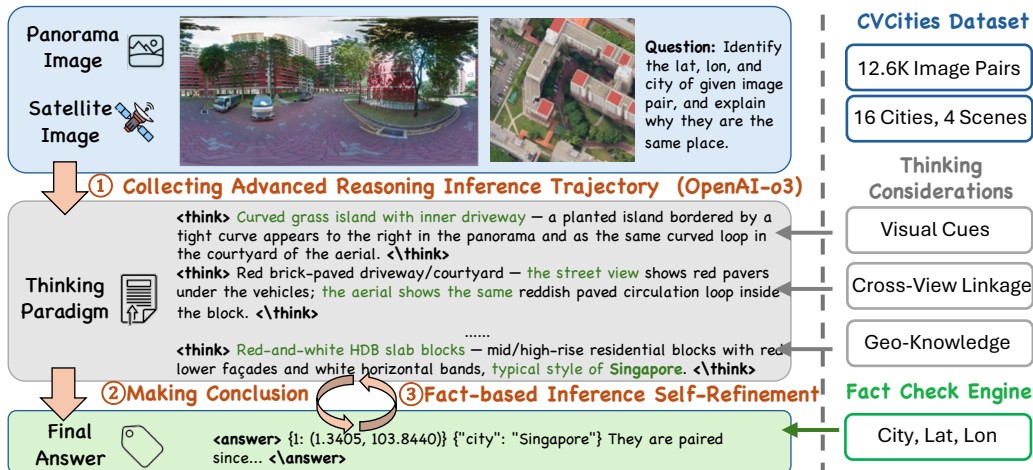

Figure 3: Geospatial thinking CoT data engine.

later benchmarks. Instead, we construct a comprehensive geospatial reasoning paradigm and use it to guide our CoT synthesis on a single multi-view reasoning task as shown in Fig. 3:

1. **Visual Cue Identification:** Systematically extract any-view geospatial and semantic features: architectural styles, vegetation/biome, road topology and markings, coastline/river patterns, topography, signage/scripts, and solar cues (sun azimuth/elevation, shadows);

2. **Knowledge Association:** Map cues to geospatial priors (climate bands, cultural/linguistic regions, urban morphology). For example, link tile roofs and narrow alleys to mediterranean Europe, or infer northern winter hemisphere from low solar elevation and shadows;

3. **Evidence Corroboration:** Cross-refer multiple, potentially weak cues across views; check consistency, resolve contradictions, and prefer hypotheses with convergent evidence;

4. **Conclusion Formulation:** Synthesize the corroborated evidence into a concise answer, optionally noting uncertainty when evidence is limited.

Our scaffolding practice differs from traditional SFT philosophy of "cold-starting" VLMs on diversified target tasks just in order to accelerate the reward acquiring process during RL. Instead, we teach the target VLM a unified geospatial reasoning paradigm, which is created using a single template and from a single data source.

Such a design shift brings in multi-fold advantages: (1) It provides similar benefits of regular SFT, including stabilizing early training of RL, enabling RL to rampup faster, and also achieving the goal of teaching generic reasoning behaviors that transfer to diverse downstream tasks; (2) Moreover, the scaffolding SFT design can greatly reduce the SFT task diversity, further leading to reduction in amount of SFT steps needed. This is a key for Geo-R1 to prevent catastrophic forgetting compared to other heavy SFT stages in common post-training frameworks. (3) Lastly, such a unified CoT data acquiring process is very efficient in geospatial domain, which involve much less QA design and collection overheads facing limited geospatial data annotation sources.

## 4.2 GEOSPATIAL COT DATA ENGINE

**CoT Synthesis.** We collect images with from cross-view geospatial dataset, CV-Cities (Huang et al., 2025), which contains 223,736 panorama-satellite image pairs with geolocation data. The samples span 16 cities across 13 countries and cover four major daily scenes (city, natural, water, occlusion).

As shown in Fig.3, we prompt OpenAI-o3[1] to produce city labels and latitude/longitude for panorama - satellite image pairs, explicitly instructing cross-view analysis following our thinking paradigms: identify salient visual cues across perspectives, perform multi-view feature matching,

---

[1] https://openai.com/index/introducing-o3-and-o4-mini/

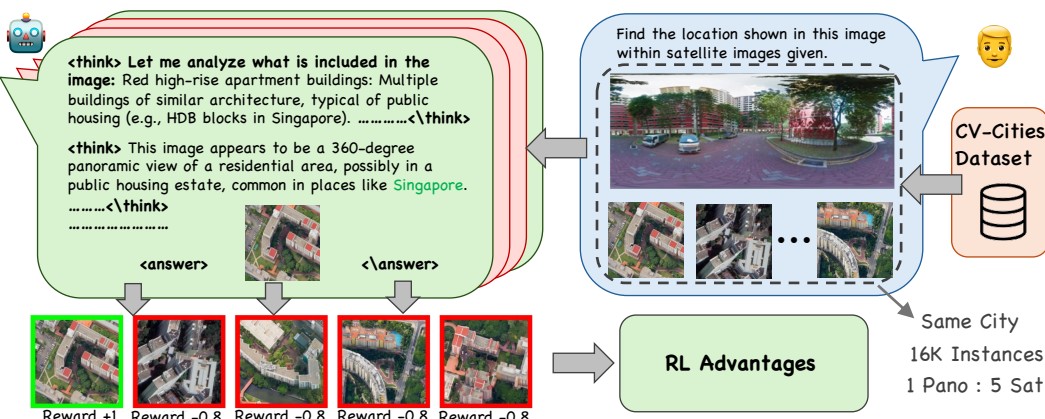

Figure 4: Cross-view pairing task for reinforcement learning with verifiable rewards.

and integrate pertinent geospatial knowledge. We collect the intermediate reasoning trajectories to form a corpus of 12,646 samples (around 43.6 MB of text). To reduce overfitting during SFT, we set to medium reasoning strengths to generate moderately detailed intermediate CoTs.

**Fact-Check Engine.** We implement the fact-check engine to serve as an automatic verifier that grounds the model's reasoning and final answers against concrete geospatial metadata. It reduces hallucinations, enforces adherence to real-world constraints, and ensures that the reasoning process remains anchored to verifiable spatial facts. The fact-check engine works through inference-based self-refinement (Madaan et al., 2023; Liu et al., 2024a): after o3 generates an initial reasoning trajectory and tentative answer, the engine cross-validates key outputs, (e.g. predicted city, latitude, and longitude) against curated factual references. If inconsistencies are detected (e.g., mismatched coordinates or unsupported city claims), the system prompts in a new conversation with both CoTs and GTs to refine the reasoning trace to be factually coherent and geographically consistent.

### 4.3 Supervised Finetuning

During the SFT training process, we fine-tune the base model (`Qwen2.5-VL-7B`, Bai et al. (2025)) with the synthetic dataset as discussed in Section 4.2. The training objective is formulated as a standard autoregressive, next-token prediction. The model is trained to minimize the cross-entropy loss over the target sequence, which is a concatenation of the CoT string (<think>... <\think>) and the answer (<answer>... <\answer>). We perform full-parameter fine-tuning on the model, including the language backbone, visual tower and multi-modal projector.

## 5 Geo-R1 Stage#2: Geospatial Thinking Elevating with RLVR

While the scaffolding stage equips the model with a structured geospatial reasoning paradigm, it does not guarantee factual precision, robustness, or multi-view consistency. To bridge this gap, the **elevating** stage leverages RLVR to improve reasoning quality under verifiable rewards.

### 5.1 Weak Supervision for Strong Elevation

The most critical challenge in RLVR lies in designing a good reward task, that can (1) be verified at scale, and (2) motivate high-quality reasoning behaviors. Achieving scalability is hard in geospatial reasoning as the only scalable supervision along with raw images is their metadata. Therefore, the key becomes how to leverage weakly-supervised metadata to create challenging reasoning tasks.

We propose the proxy RL task: matching a ground-level panoramic image to its corresponding satellite view image with confusers, as shown in Fig. 4. We let the model perform a $k$-way single-choice task. For each panoramic image $I_p$, a set of $k$ satellite image candidates $\{I_s^1, \cdots, I_s^k\}$ is provided. This set contains exactly one correct match and $k$-1 challenging confusers, such as satellite

images of nearby but incorrect locations within the same city. The model's objective is to generate high-quality reasoning to help correctly identify the choice of the matching satellite image.

Cross-view pairing is a task that is both challenging to solve and easy to verify. On the one hand, a general-purpose VLM that has not been specifically trained on cross-view pairing performs close to random guess, since satellite images selected from the same city often exhibit highly similar architectural and vegetation styles, as shown in Fig. 4. Moreover, the level of reasoning quality required for this task is extremely difficult to reach through SFT. This is proved in Table 1, for single-choice task with five options, `Qwen2.5-VL-7B` model achieves only 19% accuracy. After undergoing a phase of SFT training and being injected with substantial latent knowledge and positive CoT examples linking cross-view images, the model only gains approximately +4% in performance to 23%, barely outperforming random guesses. Therefore, such reward is nearly impossible to hack unless the model truly learns to identify useful corresponding visual cues efficiently and accurately.

On the other hand, the cross-view pairing task is easy to verify and suits large-scale RLVR training since raw images and metadata are broadly available. Such a challenging but non-hackable reward motivates the model to continuously refine its explorations and elevate the model's reasoning quality, ultimately learning strong geospatial reasoning foundation. In our experiments, we found model under-through RLVR demonstrates much stronger reasoning behaviors to capture and synthesize various types of visual information, including car plates, billboard, texts, cultural elements, tree types, traffic signs, building colors and styles, and even car brands, all serving as the foundation for any out-of-domain generalized geospatial reasoning.

## 5.2 REWARD DESIGN

To make the model's output align with our preferences for better output, we combine direct, verifiable outcome rewards with light textual shaping:

$$r = \lambda_{\mathrm{acc}}\, r_{\mathrm{acc}} + \lambda_{\mathrm{fmt}}\, r_{\mathrm{fmt}} + \lambda_{\mathrm{len}}\, r_{\mathrm{len}} + \lambda_{\mathrm{rep}}\, r_{\mathrm{rep}}.$$

**Accuracy.** $r_{\mathrm{acc}} = +1$ if the correct $\hat{I}_s = I_s^{\star}$ is identified, $-0.8$ if $\hat{I}_s \neq I_s^{\star}$, and $-1$ for non-answers / malformed choices; this yields a dense, calibrated signal while discouraging degenerate refusals.

**Format.** $r_{\mathrm{fmt}} \in \{0, 1\}$ grants credit for emitting the agreed <think>/<answer> structure.

**Length.** $r_{\mathrm{len}}$ rewards succinct but sufficient justification (encourage thinking, avoid over-thinking). We adopt cosine reward (Yeo et al., 2025) with parameters $r_0^c = 0, r_0^w = -1, r_L^c = 0.5, r_L^w = 0$, punishing overly brief responses when the correct answer is not provided. See details in Appendix A.

**Repetition.** $r_{\mathrm{rep}} \leq 0$ penalizes token-level pattern loops to mitigate post-training repetition. This mixture encourages disciplined reasoning that is both verifiable and readable.

## 5.3 REINFORCEMENT LEARNING POST-TRAINING

We optimize with Group Relative Policy Optimization (GRPO, Shao et al., 2024). For each prompt we sample $M$ rollouts $\{y^{(m)}\}_{m=1}^M$, compute rewards $\{r^{(m)}\}$, and form group-wise advantages:

$$A^{(m)} = r^{(m)} - \frac{1}{M} \sum_{j=1}^{M} r^{(j)}. \tag{1}$$

The policy $\pi_\theta$ is updated with a clipped objective and a KL regularizer:

$$\max_{\theta} \; \mathbb{E}\left[\min\left(\rho_\theta^{(m)} A^{(m)}, \; \mathrm{clip}(\rho_\theta^{(m)}, 1-\epsilon, 1+\epsilon)\, A^{(m)}\right)\right] - \beta\, \mathrm{KL}(\pi_\theta \,\|\, \pi_{\mathrm{ref}}), \tag{2}$$

where $\rho_\theta^{(m)} = \frac{\pi_\theta(y^{(m)}|x)}{\pi_{\mathrm{ref}}(y^{(m)}|x)}$ is the likelihood ratio and $\pi_{\mathrm{ref}}$ is a frozen reference model.

**Outcome-Based Reward.** We allow free-form '<think>' before parsing a single final choice in '<answer>'. The outcome reward ensures stable gradients even at small scales, while textual shaping regularizes behavior and prevents verbosity drift. In practice we (i) filter unparseable rollouts via the format check, (ii) anneal, lowering the sampling temperature during training to gradually reduce exploration, and (iii) monitor the KL term to avoid collapsing to the reference or diverging into reward-hacking.

## 6 EXPERIMENTS AND RESULTS

In this section, we present the implementation setup and experimental results of Geo-R1 during training stage and testing on standard benchmarks. Our training dynamic recording shows a geospatial "Aha Moment" (Guo et al., 2025), a signal of success of RL-based inference-time scaling. We evaluated the model's performance on both in-distribution and out-of-distribution datasets, demonstrating that during RL training, the model not only correctly learns cross-view pairing tasks but also acquires broader, generalizable geospatial reasoning capabilities.

### 6.1 SETTINGS AND IMPLEMENTATION DETAILS

We implement Geo-R1 with LLama-Factory (Zheng et al., 2024) and VLM-R1 (Shen et al., 2025), two open-source LLM post-training frameworks for fast and stable SFT and GRPO training. We use `Qwen2.5-VL-7B` (Bai et al., 2025) as the base model, and then conclude a `Geo-SFT` intermediate state model after the stage-1 scaffolding-oriented SFT. Starting from `Geo-SFT` model, we conduct the stage-2 training and get the final `Geo-R1` model. We also conduct RL training independently, starting directly from the base model, resulting in `Geo-R1-Zero`. We conduct full-parameter fine-tuning on the model for more stable convergence and higher final accuracy. We train the model on 8×NVIDIA H100 GPUs. We employ vLLM (Kwon et al., 2023) to accelerate model inference during RL and testing phases. Training details are provided in the Appendix B.

### 6.2 IN-DISTRIBUTION GEOSPATIAL REASONING

**Benchmark.** We evaluate the in-distribution cross-view pairing task. We sample 5,000 holdout sets of images from CV-Cities using the same method as described in Section 5.1, to serve as the test set. They do not overlap with the data points sampled during either the SFT stage or the RL stage.

**Remark 1: SFT Fails on Cross-View Pairing.** We found that learning cross-view pairing using only positive examples through SFT does not generalize well. As shown in Table 1, the `Geo-SFT` model can only marginally outperform the base model by 4.1%, which is still extremely close to random guess (20% accuracy). We also observe a significant increase in the completion length of the `Geo-SFT` model attributed to substantial content duplication, which indicates heavy SFT is not a good fit for complex and generalizable geospatial reasoning tasks.

**Remark 2: RL Generalizes on Cross-View Pairing.** RL delivers robust performance improvements for the Cross-View Pairing task through both positive and negative instances feedback. As shown in Table 1, both `Geo-R1` and `Geo-R1-Zero` model achieve a significant performance boost in terms of about 60% accuracy gain. This means that during the RL process, the model does not merely memorize images but learns how to distinguish between images from multiple viewpoints.

Regarding completion length, thanks to the length reward and repetition penalty, the inference completion length of `Geo-R1` and `Geo-R1-Zero` is kept within a reasonable range, avoiding the excessive repetition seen in `Geo-SFT`. Benefiting from the thinking paradigm learned during the scaffolding SFT phase, `Geo-R1` exhibits a more concise and regular intermediate reasoning process compared to `Geo-R1-Zero`, demonstrated by a completion length approximately 1/3 shorter.

### 6.3 OUT-OF-DISTRIBUTION GEOSPATIAL REASONING

A key contribution of Geo-R1 is to unlock the open-ended geospatial reasoning capability of VLM. Notably, we evaluate the `Geo-R1`'s performance on OOD datasets all under *zero-shot settings*.

#### 6.3.1 STREET VIEW GEOCHAIN

**Benchmark.** GeoChain (Yerramilli et al., 2025) is a geospatial reasoning benchmark which employ template-based chain-of-thought to solve a geolocation task across 20 cities, of which 15 cities are

Table 1: Results on in-distribution cross-view pairing task.

|  | Qwen2.5-VL-7B (Base Model) | Geo-SFT | Geo-R1-Zero | Geo-R1 |
|---|---|---|---|---|
| **Accuracy** | 19.0% | 23.1% | 78.1% | **82.4%** |
| **Completion Length** | 204.6 | 1127.6 | 587.4 | 378.8 |

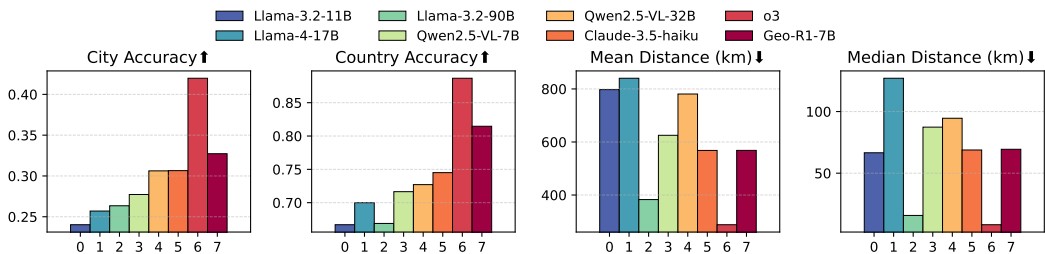

Figure 5: Results on IMAGEO dataset-GSS (Li et al., 2025)

Table 2: Comparison with task-specific geo-localization model on IMAGEO-GSS dataset.

| model | city_accuracy | country_accuracy | mean_distance_km | median_distance_km |
|---|---|---|---|---|
| GeoCLIP | 0.1086 | 0.6361 | 943.4838 | 266.8987 |
| Geo-R1 | 0.3272 | 0.8146 | 568.3228 | 69.4008 |

OOD. We did not use their CoT template but allow free-from reasoning of our models to answer the questions. We evaluated the model's performance on 13 subproblems with explicit ground-truth data. See GeoChain problem details in Appendix D.

**Remark 3: SFT + RL Generalizes Best for Complex OOD Reasoning.** As shown in Fig. 1, across all tasks spanning geographical, enviromental to cultural ones, `Geo-R1` consistently outperforms both the base model and intermediate models. The comparison with the `Geo-SFT` model and `Geo-R1-Zero` model demonstrates that both the Scaffolding and Elevating phases are indispensable. Harmonizing both yileds the best performance for geospatial reasoning.

### 6.3.2 STREET VIEW IMAGEO-BENCH

**Benchmark.** The IMAGEO-Bench Li et al. (2025) is a systematic OOD benchmark that evaluates large language models' ability to perform image geolocalization by testing accuracy, distance error across diverse datasets of global (6152 images from 396 cities) and US (2928 images).

**Remark 4: Geo-R1 Outperforms Open-Source LLMs.** As shown in Fig. 5, we evaluated multiple open-source and closed-source models on the IMAGEO-GSS dataset. Our results show that Geo-R1 achieves the highest city and country identification accuracy among all open-source models. Note `Llama-3.2-90B` (Dubey et al., 2024) appears lower mean and median distance since they calculate it on top of successful responses only (success rate of 46%), while ours successful response rate are 99%. The close-source `o3` continues to hold an absolute lead in this benchmark, which we attribute to its tremendous parameter scale and reinforcement learning efforts. We include more details, including latitude, longitude analysis in Appendix E.

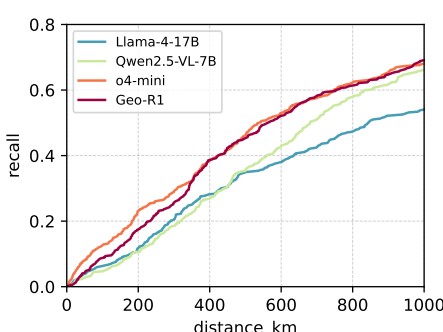

Figure 6: Results on RSTeller geolocation.

**Remark 5: Geo-R1 Outperforms task-specific geo-localization models.** We compare our model with Non-LLM domain-specific models targeting geo-localization task such as GeoCLIP (Vivanco Cepeda et al., 2023). As shown in Table 2, on scene-diverse IMAGEO-GSS dataset, `Geo-R1` surpasses GeoCLIP in all aspects of city-scale and country-scale accuracy, as well as in its prediction error. This indicates that `Geo-R1` has demonstrated its robustness across diverse weather conditions, scenarios, and environments. Furthermore, we compared our model against a series of task-specific models trained specifically for IM2GPS on geolocation's commonly used IM2GPS3K benchmark. As shown in Table 3, we tested `Geo-R1` in two modes: (1) without any thinking, directly outputting answers (w/o thinking), and (2) enabling thinking mode before providing responses (w/ thinking). In both modes, the model operates entirely in zero-shot mode. When enabled in thinking mode, the

Table 3: Performance comparison across different geographic localization thresholds.

| Method | Street 1 km | City 25 km | Region 200 km | Country 750 km | Continent 2500 km |
|---|---|---|---|---|---|
| [L]kNN, $\sigma = 4$ (Vo et al., 2017) | 7.2 | 19.4 | 26.9 | 38.9 | 55.9 |
| PlaNet (Weyand et al., 2016) | 8.5 | 24.8 | 34.3 | 48.4 | 64.6 |
| CPlaNet (Seo et al., 2018) | 10.2 | 26.5 | 34.6 | 48.6 | 64.6 |
| ISNs (Muller-Budack et al., 2018) | 10.5 | 28.0 | 36.6 | 49.7 | 66.0 |
| Translocator (Pramanick et al., 2022) | 11.8 | 31.1 | 46.7 | 58.9 | 80.1 |
| GeoDecoder (Clark et al., 2023) | 12.8 | 33.5 | 45.9 | 61.0 | 76.1 |
| GeoCLIP (Vivanco Cepeda et al., 2023) | 14.1 | 34.5 | 50.6 | 69.7 | 83.8 |
| Qwen2.5-VL-7B-Instruct | 5.0 | 26.0 | 46.1 | 64.8 | 77.5 |
| Geo-R1 (w/o. thinking, zero-shot) | 13.2 | 27.2 | 49.0 | 68.2 | 81.4 |
| Geo-R1 (w/. thinking, zero-shot) | **16.2** | **41.3** | **55.4** | **78.1** | **88.9** |

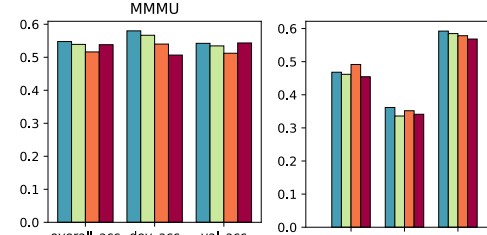 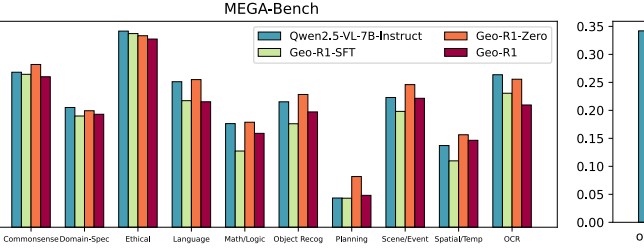

Figure 7: Results on non-geospatial general-purpose task benchmarks.

model achieves state-of-the-art performance. `Geo-R1`'s direct output performance is also competitive without thinking. This fully demonstrates that Geo-R1's geolocation capabilities are based on the model's fundamental understanding of geospatial information, rather than overfitting to data.

### 6.3.3 SATELLITE VIEW GEOLOCATION

**Remark 6: Geo-R1 Generalizes Better on OOD RSTeller Geolocation.** We further consider an out of distribution geolocation task: estimating the location of high-resolution aerial images. We adopt a subset of RSTeller (Ge et al., 2025), which is a data distribution (U.S. Agricultural Land) `Geo-R1` has not encountered before. As shown in Fig. 6, `Geo-R1` achieves higher recall than the base model and is on-par with `o4-mini`, indicating that our post-training approach demonstrates generalization on new OOD tasks. See Appendix F for more details.

### 6.4 PRESERVATION OF PRIMITIVE ABILITIES

**Remark 7: Geo-R1 Avoids Catastrophic Forgetting.** We find that our post training does not noticeably decrease the performance on the primitive tasks. We evaluate `Geo-R1`, the base model `Qwen2.5-VL-7B`, as well as `Geo-R1-Zero` and `Geo-SFT` on general purpose VLM benchmarks like MEGA-Bench (Chen et al., 2025), GPQA (Rein et al., 2024), and MMMU (Yue et al., 2024). As shown in Fig. 7, `Geo-R1` effectively preserves the base model's capabilities in scientific QA, foundational multimodal understanding, etc. See Appendix G for more details.

Notably, we can observe most slight performance degradation on primitive tasks are brought by SFT (green bar), but not by RLVR (orange bar). This indicates the necessity to carefully control SFT steps to be small to avoid catastrophic forgetting. This also highlights our scaffold SFT's advantages to achieve a good tradeoff to use minimal SFT steps for geospatial reasoning paradigm learning.

### 6.5 TRAINING DYNAMICS

By observing the model's training dynamics, we identified several noteworthy phenomena, which we remark in this section. We describe the model's training dynamics in detail in the Appendix C.

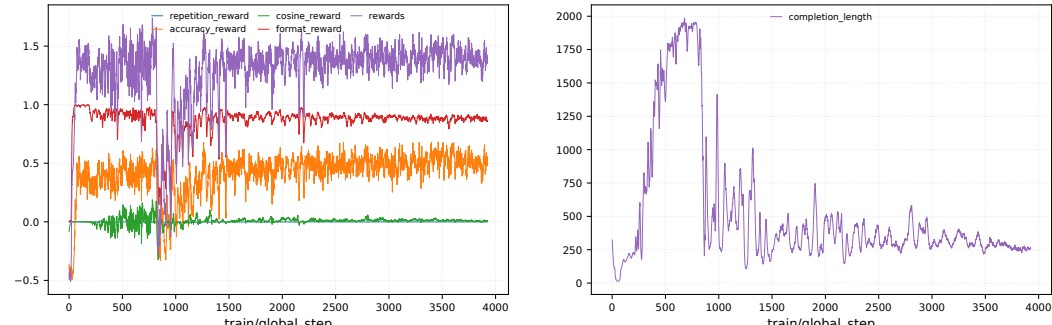

Figure 8: GRPO training dynamics. **Left:** rewards dynamic. **Right:** completion length.

Table 4: Generalization ability of Geo-R1 framework test on IMAGEO-GSS dataset.

| model | city_accuracy | country_accuracy | mean_distance_km | median_distance_km |
|---|---|---|---|---|
| InternVL3-8B | 0.146229 | 0.511513 | 1352.624585 | 357.089196 |
| + Geo-R1 | 0.312576 | 0.723197 | 603.157342 | 84.312461 |
| Qwen2.5-VL-7B | 0.277271 | 0.716517 | 625.207248 | 87.365986 |
| + Geo-R1 | 0.327264 | 0.814664 | 568.322859 | 69.400873 |

**Remark 8: Geospatial "Aha Moment".** During the RL training, as shown in Fig. 8, we observed that the model's reward reached its first peak around 100 steps. We observe that the model's completion length does not exhibit a convergence trend consistent with the reward over the subsequent period. The model's completion length exhibits a pattern of first decreasing and then increasing, consistent with the "Aha moment" observed in Deepseek-R1 (Guo et al., 2025), while the model accuracy reward continues to rise till convergence. We refer to this as the geospatial "Aha-Moment".

**Remark 9: Outputs Stabilize after Double Ascents.** We observe that the model's behavior stabilizes after two ascents. That is saying, at the begining of the RL training, as the model trained, its outputs became increasingly longer but unstable. The model tends to engage in extensive deliberation, but the content of its deliberations is meaningless or redundant. Then, as shown in Fig. 8 the model's reward collapses after exceeding the maximum output length limit (2048). The model is further trained over the subsequent 500 steps until convergence. We find that the model no longer hit the completion length wall. Meanwhile, the model developed a stable and effective intermediate reasoning process during this double ascents of the rewards. We show some examples in Appendix C.

## 6.6 GEO-R1 FRAMEWORK GENERALIZATION

Remark 10: Geo-R1 is Architecture-Agnostic. To further verify the universality of our proposed framework, we applied the Geo-R1 post-training paradigm to a different VLM architecture, `InternVL3-8B` (Zhu et al., 2025), and evaluated it on the out-of-distribution IMAGEO-GSS dataset. As reported in Table 4, Geo-R1 yields substantial improvements regardless of the backbone: the post-trained `InternVL3-8B` achieves a city accuracy of 31.26%, more than doubling the base model's 14.62%, while drastically reducing the median distance error from 357.09 km to 84.31 km. While for the in-distribution cross-view pairing task, the Geo-R1 framework improves the `InternVL3-8B`'s accuracy from 20.3% to 84.7%. This demonstrates that Geo-R1 can consistently unlock latent geospatial reasoning capabilities and achieve performance gain across diverse VLM architectures.

## 7 CONCLUSION

In this work, we introduced Geo-R1, a reasoning-centric post-training framework that harmonizes supervised fine-tuning and reinforcement learning to unlock advanced geospatial inference in vision–language models. Our results demonstrate that Geo-R1 achieves substantial gains on both in-distribution and out-of-distribution geospatial tasks. Geo-R1 highlights the promise of reasoning-first post-training as a scalable path toward robust and generalizable geospatial intelligence.

## REPRODUCIBILITY STATEMENT

We are dedicated to developing open-source models. Our code is available at `https://anonymous.4open.science/r/Geo-R1-ICLR`. The data generation method is described in detail in Sec. 4, and the relevant prompts can be found in the Appendix B. We commit to open-sourcing the model weights for testing and continuing development.

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

## A    COSINE REWARDS

We use a cosine-shaped length reward (Yeo et al., 2025) to encourage *succinct but sufficient* reasoning: reward increases smoothly with the number of generated reasoning tokens until a target cap and then plateaus (no incentive to over-think).

**Setup.**    Let $n \in \mathbb{N}$ be the number of reasoning tokens produced before the final answer token. Let $n_{\min} \geq 0$ be the minimum length after which we start rewarding, and $n_L > n_{\min}$ be the target cap beyond which extra tokens bring no additional length reward. Define the normalized clipped length

$$s(n) = \text{clip}\left( \frac{n - n_{\min}}{n_L - n_{\min}}, \, 0, \, 1 \right), \quad \phi(s) = \tfrac{1}{2}(1 - \cos(\pi s)) \in [0, 1].$$

Here $\phi(\cdot)$ is monotonically increasing, has zero slope at both ends, and provides a smooth rise without sharp incentives to chase the cap, we set the $n_{min} = 0$ and $n_L = 2048$ in the Geo-R1 training.

Let $y \in \{c, w\}$ denote whether the final answer is correct ($c$) or wrong ($w$). Given boundary rewards $\{r_0^y, r_L^y\}$ (at $s = 0$ and $s = 1$ respectively), the cosine length reward is

$$r_{\text{len}}(n, y) = r_0^y + \left( r_L^y - r_0^y \right) \phi\big(s(n)\big). \tag{A.1}$$

Intuitively, $r_0^y$ controls how we treat very brief responses, while $r_L^y$ sets the maximum bonus once a sufficient justification is reached.

**Instantiated parameters.**    In our Geo-R1 training case, we use

$$r_0^c = 0, \qquad r_0^w = -1, \qquad r_L^c = 0.5, \qquad r_L^w = 0. \tag{A.2}$$

Thus, (i) a very short response that is *wrong* receives a negative signal ($-1$), penalizing "guessing and quitting"; (ii) once sufficient length is reached, a *correct* response gets a modest bonus ($+0.5$), while a *wrong* response receives no additional bonus, avoiding incentives to "pad" incorrect reasoning; and (iii) beyond $n_L$ there is no further gain, discouraging over-thinking.

Equation equation A.1 is an episodic scalar reward added to other task terms (e.g., accuracy, format). Let $\lambda_{\text{len}} \geq 0$ be a weight; the total reward is

$$R = R_{\text{task}} + \lambda_{\text{len}} \, r_{\text{len}}(n, y).$$

We tune $\lambda_{\text{len}}$ on held-out tasks; $n_{\min}$ and $n_L$ are hyperparameters tied to the allowed rationale budget (e.g., $n_{\min}$ for ignoring boilerplate, $n_L$ near the per-sample token cap).

**Properties.**    (i) **Monotone & bounded:** $r_{\text{len}}$ increases smoothly from $r_0^y$ to $r_L^y$ as $n$ grows from $n_{\min}$ to $n_L$ and is constant thereafter.    (ii) **Short-penalty asymmetry:** with equation A.1–equation A.1 we penalize short *wrong* answers while not penalizing short *correct* ones, aligning incentives toward concise correctness. (iii) **No incentive to pad:** because $\phi(1) = 1$ and is flat beyond $n_L$, longer-than-needed rationales do not increase reward.

## B    TRAINING DETAILS

### B.1    SUPERVISED FINE-TUNING

We use the LLama-Factory (Zheng et al., 2024) for the supervised fine-tuning. We first conduct supervised fine-tuning on the multimodal backbone using the `Qwen/Qwen2.5-VL-7B-Instruct` (Bai et al., 2025) model as initialization. The model is trained in full fine-tuning mode without freezing any modality-specific components. The maximum input sequence length is set to 131,072 tokens, and up to 10M samples are used for training. Optimization is performed with a cosine learning rate scheduler, peak learning rate of $1.0 \times 10^{-6}$, and warmup ratio of 0.1. Each GPU processes a batch size of 1, and we accumulate gradients for 2 steps. Training is conducted for 2 epochs with bfloat16 precision. Key hyperparameters are summarized in Table 5.

Table 5: Key training hyperparameters in SFT stage of Geo-R1.

| Parameter | Value |
|---|---|
| Fine-tuning type | Full |
| Max input length | 131072 |
| Max samples | 10M |
| Batch size (per device) | 1 |
| Gradient accumulation steps | 2 |
| Learning rate | $1.0 \times 10^{-6}$ |
| Epochs | 2.0 |
| Scheduler | Cosine |
| Warmup ratio | 0.1 |
| Precision | bfloat16 |
| DeepSpeed Config | ZeRO-2 |
| Freeze Vision Tower | False |
| Freeze Multi-Modal Projector | False |

## B.2 GRPO-BASED REINFORCEMENT LEARNING

After SFT, we further optimize the model using Group Relative Policy Optimization (Shao et al., 2024). We employ the VLM-R1 (Shen et al., 2025) as the training framework. Training is launched with `torchrun` on 8 A100 GPUs (single node). We employ DeepSpeed ZeRO-3 for memory-efficient distributed optimization. Each GPU uses a per-device batch size of 4, with gradient accumulation of 2 steps, yielding an effective batch size of $4 \times 2 \times 8 = 64$ prompts per update. For each prompt, the model generates 8 candidate completions, resulting in 512 generations per update. The maximum completion length is set to 2048 tokens. Reward functions include `accuracy`, `format`, `length`, and `repetition`, with a KL/entropy regularization coefficient $\beta = 0.04$. We adopt FlashAttention-2, gradient checkpointing, and mixed precision (bfloat16) to improve efficiency. GRPO-specific hyperparameters are summarized in Table 6, 7, and 8.

Table 6: System and parallel configuration for GRPO training.

| Item | Setting |
|---|---|
| GPUs per node | 4/8 |
| Nodes | 1 |
| Total GPUs | 4/8 |
| Precision | bfloat16 |
| Attention kernel | FlashAttention-2 |
| Gradient checkpointing | Enabled |
| DeepSpeed Config | ZeRO-3 |

Table 7: Training schedule and bookkeeping.

| Item | Setting |
|---|---|
| Epochs | 2 |
| Per-device batch size | 4 |
| Gradient accumulation | 2 |
| *Effective prompt batch / update* | $4 \times 2 \times 8 = 64$ |
| Logging interval | 1 |
| Max completion length | 2048 tokens |

Table 8: GRPO-specific configuration.

| Item | Setting |
|---|---|
| Generations per prompt | 8 |
| *Total generations / update* | $64 \times 8 = 512$ |
| Reward functions | `accuracy, format, length, repetition` |
| KL/entropy coefficient | $\beta = 0.04$ |

During the RL phase, we adopt the following system prompt:

```
"A conversation between User and Assistant. The user asks a
    question, and the Assistant solves it. The assistant first
    thinks about the reasoning process in the mind and then provides
    the user with the answer. The reasoning process and answer are
    enclosed within <think> </think> and <answer> </answer> tags,
    respectively, i.e., <think> reasoning process here
    </think><answer> answer here </answer>"
```

A data sample is defined as:

```
{"id": 1, "image":
    ["cv_cities_16k/barcelona/pano_img/--0eE3ZmREVxVXH_oIeIqw.jpg",
    "cv_cities_16k/barcelona/sat_img/--0eE3ZmREVxVXH_oIeIqw.jpg",
    "cv_cities_16k/barcelona/sat_img/1hfQgX1jGYsXaP74MfLSKQ.jpg",
    "cv_cities_16k/barcelona/sat_img/1plY2fbvDkM9yadGq_edzw.jpg",
    "cv_cities_16k/barcelona/sat_img/-cM5TsqoZcV-kY1xxOARBA.jpg",
    "cv_cities_16k/barcelona/sat_img/2iA9_BNIeO3XZgbLamEbPA.jpg"],
    "conversations": [{"from": "human", "value":
    "<image><image><image><image><image><image> You are shown one
    ground-level panorama and five satellite views labeled as A, B, C,
    D, and E. Exactly one satellite image depicts the same location.
    Identify the correct satellite image. Think step by step, you can
    generate multi <think> </think> box, bound your each thinking step
    with a box. Respond with a single choice A-E in <answer>
    </anwser>."}, {"from": "gpt", "value": "A"}]}
```

## C    TRANING DYNAMICS

We show here the policy evolution during GRPO training, aligning with the quantitative trends shown in Figs. 9-18. We report the overall return and dispersion, component-wise rewards (accuracy, repetition, format, and length), optimization diagnostics (loss and gradient norms), and the behavior of completion lengths.

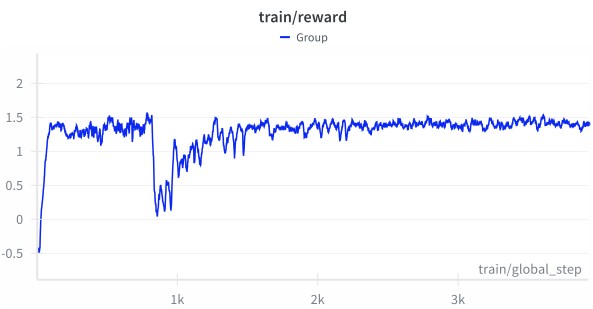

Figure 9: Reward dynamic during GRPO training.

### C.1    OVERALL REWARD AND DISPERSION

**Overall reward.** As shown in Fig. 9, We observe a rapid rise in average reward at the beginning, followed by a brief stabilization, a secondary climb, and then a steady plateau. The first prominent peak appears within the early updates and matches the "geospatial *aha-moment*" described in the main text: the policy starts to assemble consistently useful spatial cues before settling into a higher-reward regime.

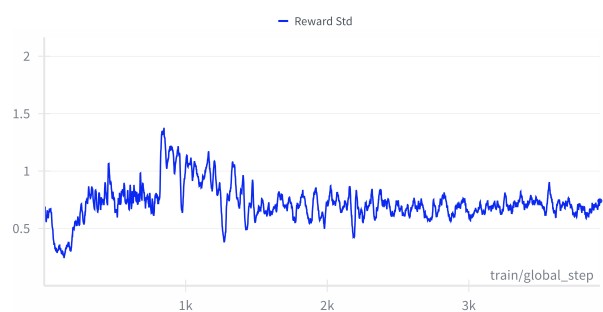

Figure 10: Reward standard deviation dynamic during GRPO training.

**Reward dispersion. (Fig. 10)** The within-batch standard deviation is high in the exploratory phase—reflecting diverse and unstable reasoning paths—and gradually contracts as decoding temperature is annealed and format filtering becomes effective. Short, local upticks in variance coincide with exploration boosts or scheduler changes.

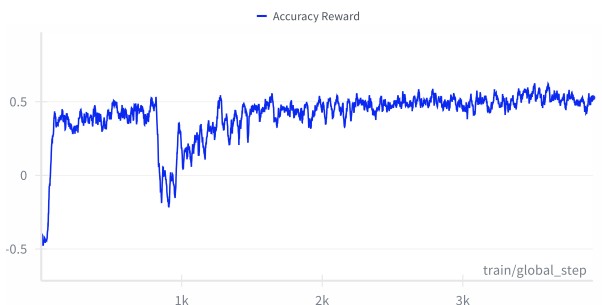

Figure 11: Accuracy reward dynamic during GRPO training.

### C.2 COMPONENT-WISE REWARDS

**Accuracy reward ($r_{\mathrm{acc}}$).** The mean of $r_{\mathrm{acc}}$ increases monotonically and saturates near the end of training (Fig. 11). We assign a positive credit to correct predictions and a negative credit to incorrect or unparseable outputs, which gives a dense, calibrated learning signal while discouraging "no-answer" degeneracy.

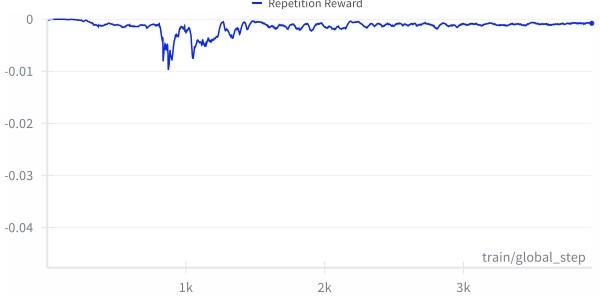

Figure 12: Repetition reward dynamic during GRPO training.

**Repetition reward ($r_{\mathrm{rep}} \leq 0$).** The magnitude of the repetition penalty declines toward zero over time, indicating that the policy sheds looped phrases and mechanical echoing, and converges to more concise chains of thought. (Fig. 12)

**Format reward ($r_{\mathrm{fmt}} \in \{0, 1\}$).** The fraction of format-compliant generations rises quickly to near-saturation (Fig. 13) once the `<think>...</think><answer>...</answer>` structure is enforced. This stabilizes parsing and downstream evaluation and reduces label noise from ill-formed outputs.

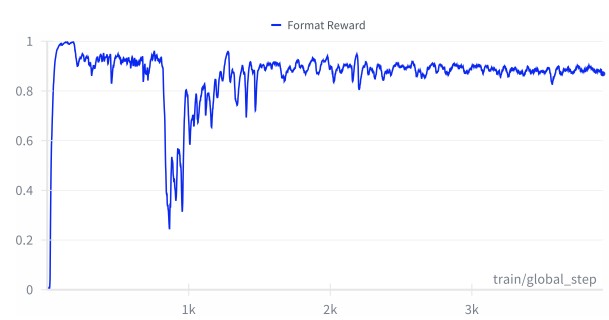

Figure 13: Format reward dynamic during GRPO training.

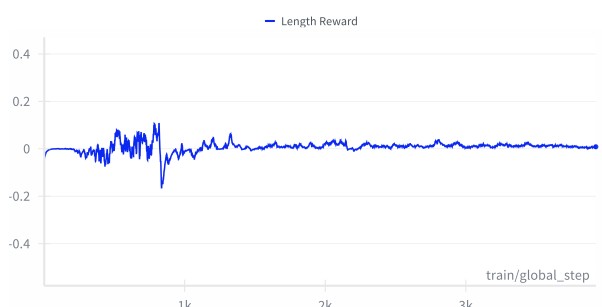

Figure 14: Length (cosine) reward dynamic during GRPO training.

**Length / cosine reward ($r_{\text{len}}$).** Empirically (Fig. 14), $r_{\text{len}}$ increases early, then plateaus; when the policy temporarily over-extends to the cap, the net return can dip, prompting a stable reversion to concise-but-sufficient chains.

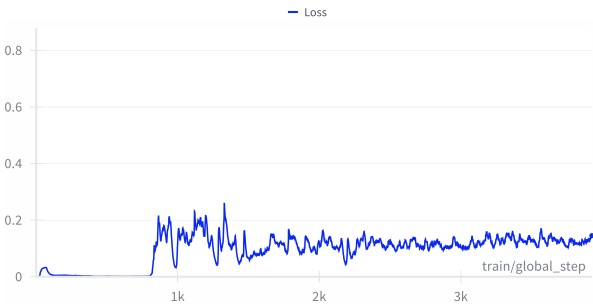

Figure 15: Loss dynamic during GRPO training.

### C.3 OPTIMIZATION DIAGNOSTICS

**Loss (Fig. 15).** The training loss decreases and then stabilizes, indicating that the policy does not exploit spurious reward loopholes but instead converges around the reference policy under the KL constraint.

**KL Divergence (Fig. 16).** KL divergence fluctuated after encountering the completion length wall and subsequently remained stable, indicating that the model has undergone certain changes relative to the original distribution, but overall remains within a controllable range.

**Gradient norm (Fig. 17).** We observe several early spikes (coinciding with shifts in accuracy/length/format trade-offs), followed by clear stabilization. In practice, large-batch sampling with efficient memory partitioning (e.g., ZeRO) and fast attention kernels keep updates well-behaved.

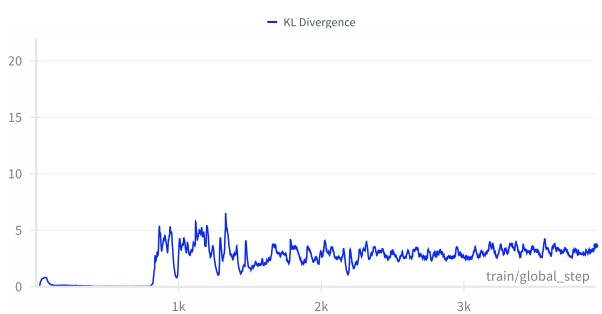

Figure 16: KL Divergence dynamic during GRPO training.

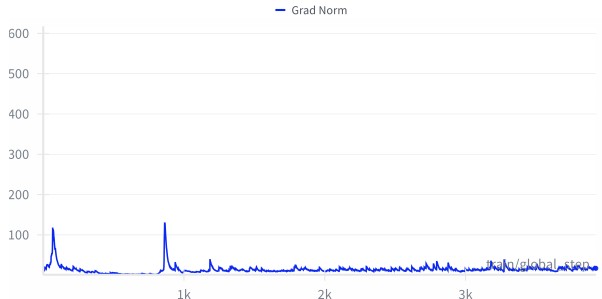

Figure 17: Gradient Norm dynamic during GRPO training.

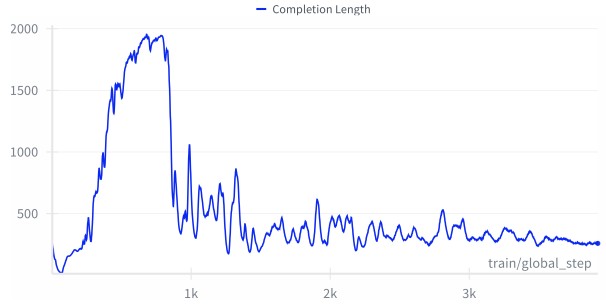

Figure 18: Completion length dynamic during GRPO training.

### C.4 COMPLETION LENGTH BEHAVIOR (FIG. 18)

Completion lengths follow a "grow → touch-cap → recede → stabilize" trajectory. In the exploratory phase, the model often hits the 2048-token limit, which, combined with the length/repetition shaping, lowers net returns and nudges the policy toward more compact and more accurate solutions. The stabilized regime features shorter completions that correlate with higher accuracy and lower dispersion.

## D GEOCHAIN RESULTS

The GeoChain (Yerramilli et al., 2025) dataset is a large-scale benchmark designed to evaluate step-by-step geographic reasoning in multimodal large language models (MLLMs). Built on 1.46 million Mapillary street-level images, it pairs each image with a 21-step chain-of-thought (CoT) sequence, resulting in over 30 million question–answer pairs. These questions progressively guide models from coarse reasoning (e.g., hemisphere, continent) to fine-grained tasks such as city-level identification and predicting precise latitude–longitude coordinates. To support detailed analysis, the dataset includes semantic segmentation maps with 150 visual classes and a locatability score that quantifies how identifiable a location is from visual cues, allowing images to be stratified into Easy, Medium, and Hard difficulty tiers. A curated subset, GeoChain Test-Mini, contains 2,088 diverse and high-quality images for focused evaluation. Overall, GeoChain provides a structured,

Table 9: Selected questions from GeoChain.

| Index | Question | Category | Difficulty |
|---|---|---|---|
| 0 | Would you say this location is near the Equator? | Geographical | Easy |
| 1 | Does this location seem to be close to the Poles? | Geographical | Easy |
| 2 | Is this place located in the Northern Hemisphere? | Geographical | Easy |
| 3 | Which continent best describes where this location is? (7 continents: North America/South America/Europe/Africa/Asia/Oceania/Antarctica) | Geographical | Easy |
| 4 | Is this place near coast? | Terrain/Environmental | Medium |
| 5 | Does this location appear to be an island? | Terrain/Environmental | Medium |
| 6 | Is this place located in a desert region? | Terrain/Environmental | Easy |
| 7 | Does this location seem to be in a mountainous or hilly region? | Terrain/Environmental | Easy |
| 8 | Does this place look like a big city? | Sociocultural | Easy |
| 9 | Would you classify this place as a small town? | Sociocultural | Medium |
| 10 | What language(s) are most likely spoken at this place? | Sociocultural | Hard |
| 11 | Can you name the state or province this place belongs to? | Geolocation | Hard |
| 12 | What is the name of the city, town, or village seen here? | Geolocation | Hard |

diagnostic framework that highlights model strengths and weaknesses across visual, spatial, cultural, and geolocation reasoning categories.

We selected 13 subproblems from GeoChain to validate the model's geospatial performance. Because these 13 questions have high-quality annotations. The description of these questions can be seen in Table 9. The subproblems in this dataset are highly challenging. We extract a subset of 800 volumes to validate the model's accuracy on these problems. The results are shown in Table 10 and Fig. 1.

Table 10: Results on GeoChain subproblems.

| Index | Qwen2.5-VL-7B | Geo-SFT | Geo-R1-Zero | Geo-R1 |
|---|---|---|---|---|
| 0 | 86.75 | 85.75 | 91.75 | 91.50 |
| 1 | 73.00 | 82.50 | 93.50 | 98.875 |
| 2 | 55.75 | 65.75 | 87.75 | 97.75 |
| 3 | 83.75 | 82.75 | 81.75 | 98.125 |
| 4 | 57.25 | 59.25 | 63.25 | 64.75 |
| 5 | 82.50 | 81.50 | 99.00 | 100.00 |
| 6 | 83.25 | 81.25 | 90.25 | 92.00 |
| 7 | 94.25 | 91.25 | 84.25 | 96.75 |
| 8 | 6.625 | 13.625 | 31.625 | 40.25 |
| 9 | 61.50 | 63.50 | 22.50 | 77.75 |
| 10 | 5.50 | 6.50 | 41.50 | 67.375 |
| 11 | 6.25 | 13.25 | 43.25 | 57.75 |
| 12 | 4.50 | 15.50 | 25.625 | 64.75 |

# E   IMAGEO-BENCH RESULTS

IMAGEO-Bench is a standardized benchmark for image geolocalization with vision-language models that emphasizes transparency, structure, and real-world diversity. It unifies input–output format via a constrained JSON schema requiring step-by-step visual reasoning (evidence from landmarks, text/signage, cultural cues, and spatial context) together with a predicted address, latitude/longitude, and confidence. The suite spans three complementary datasets—a globally distributed street-level set, a U.S. points-of-interest set, and a private held-out collection—covering outdoor/indoor scenes, urban–suburban variety, and broad geographic coverage to probe generalization and bias. The protocol disallows external tools and embedded GPS during inference to ensure comparability, and it provides reproducible scripts plus multi-granularity metrics (parsability, country/state/city correctness, and great-circle distance) alongside efficiency reporting (token usage/cost). Together,

IMAGEO-Bench offers an interpretable, diagnostics-friendly testbed for studying how models extract geospatial cues and where they succeed or fail to generalize across regions and scene types.

We tested the model's comprehensive geolocation reasoning capabilities on two datasets within IMAGEO-Bench: the global dataset-GSS with 6152 samples and the U.S.-wide dataset-UPC with 2928 samples.

Table 11: Test results on IMAGEO-GSS dataset.

| model | city_accuracy | country_accuracy | mean_distance_km | median_distance_km |
|---|---|---|---|---|
| Llama-3.2-11B | 0.140930 | 0.666941 | 797.432896 | 66.563253 |
| Llama-4-17B | 0.256990 | 0.699935 | 840.291329 | 127.015201 |
| Llama-3.2-90B | 0.263459 | 0.668849 | 382.892011 | 15.740222 |
| Qwen2.5-VL-7B | 0.277271 | 0.716517 | 625.207248 | 87.365986 |
| Qwen2.5-VL-32B | 0.306242 | 0.727081 | 780.872273 | 94.564417 |
| Claude-3.5-haiku | 0.306525 | 0.745076 | 568.169894 | 68.827582 |
| o3 | 0.419769 | 0.886685 | 288.075326 | 8.207232 |
| GPT-4o | 0.278131 | 0.786421 | 683.965141 | 88.356123 |
| InternVL3-8B | 0.146229 | 0.511513 | 1352.624585 | 357.089196 |
| Geo-R1 | 0.327264 | 0.814664 | 568.322859 | 69.400873 |

As shown in Table 11 and Table 12, Geo-R1 achieves state-of-the-art performance among open-source models on both global-scale and US-scale geolocation tasks. The Geo-R1 model with 7 billion parameters can even outperform models with 90 billion parameters. We observed that `Llama-3.2-90B` exhibits exceptionally strong coordinate prediction capabilities. This is attributed to its extremely high refusal rate, where it often declines to provide answers for uncertain queries. Consequently, the number of usable responses parsed is minimal, which we do not consider desirable.

The accuracy rates reported in in Table 11 and Table 12 are based on all identifiable responses. Geo-R1 achieved an identification success rate exceeding 99%. This implies that the actual performance gap between Geo-R1 and other open-source LLMs is significantly larger than what the IMAGEO Benchmark data reveals, particularly considering that Llama-3.2-90B only responded to instructions in about 46% of cases..

As shown in Fig 19, 20, 21, 22, Geo-R1 generally exhibits higher confidence in its own answers. We observe that the 32B model of Qwen2.5-VL demonstrates stronger benchmark performance than the 7B model, suggesting that training larger benchmark models using the Geo-R1 framework may yield a more robust geospatial reasoning model.

## F  AERIAL IMAGE GEOLOCATION

As an additonal OOD task, we consider aerial image geolocation. While there exists extensive ground view and cross-view (ground view + satellite or aerial) geolocation literature, there are no current benchmarks for geolocating aerial images. For our evaluation we use a small subset of US National Agriculture Imagery Program (NAIP) aerial imagery (490 images total) from (Ge et al., 2025). The aerial images are drawn evenly from across the United States. The image resolution is $448 \times 448$, with a ground sample distance of 0.6 meter per pixel. See Fig. 23 for some examples of the NAIP images used in our evaluation. Many of the images are very challenging, and we did not expect the models to achieve high accuracy at small range.

We employ a CoT prompt to elicit image geolocations from the VLMs. The same prompt is used to evaluate all models:

```
You are shown one aerial image. Provide your best guess of the location
    on Earth depicted by the image. Think step by step, you can generate
    multi <think> </think> box, bound each thinking step with a box.
    Respond with your answer in (latitude, longitude) coordinate tuple,
    accurate to 4 decimal places in <answer> </answer>. i.e. <answer>
    (lat, lon) </answer>.
```

Table 12: Test results on IMAGEO-UPC dataset.

| model | city_accuracy | state_accuracy | mean_distance_km | median_distance_km |
|---|---|---|---|---|
| Llama-3.2-11B | 0.033194 | 0.189310 | 955.537907 | 353.217494 |
| Llama-4-17B | 0.090444 | 0.248175 | 1217.486267 | 534.276735 |
| Llama-3.2-90B | 0.108540 | 0.239756 | 706.838244 | 162.954925 |
| Qwen2.5-VL-7B | 0.070673 | 0.185478 | 1411.940635 | 862.672667 |
| Qwen2.5-VL-32B | 0.083333 | 0.221610 | 1163.978083 | 775.544569 |
| Claude-3.5-haiku | 0.082572 | 0.300048 | 697.114125 | 258.685726 |
| o3 | 0.239331 | 0.457645 | 662.684007 | 214.273640 |
| Geo-R1 | 0.101602 | 0.284631 | 840.645108 | 468.950354 |

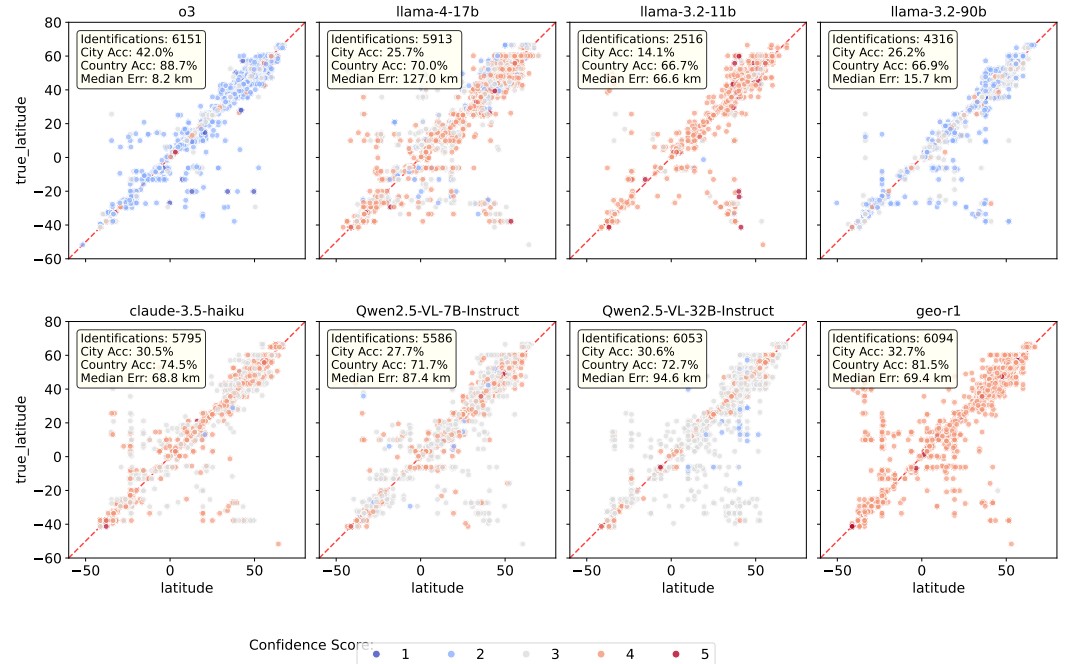

Figure 19: Latitude analysis of IMAGEO-GSS

Given the model response location and ground truth, we calculate the great-circle distance (Haversine) in kilometers, to obtain geolocation error. In our configuration, the models can return image locations from anywhere on Earth. We consider recall for distances less than 1000 km, as considering larger ranges is not practical on a US scale.

We show the recall rate at different distances. As shown in Fig. 6 and Table 13, we show that our `Geo-R1` model can on-par the advanced close-source reasoning model `o4-mini`. Our model achieved significantly better performance than `Llama-4-17B` and the base model.

Table 13: Model recall as a function of great circle distance threshold, for small subset of RSTeller aerial data (474 images).

| Method | 1 km | 25 km | 200 km | 750 km | 2500 km |
|---|---|---|---|---|---|
| GPT-o4-Mini | 0.0 | 4.6 | 23.0 | 60.2 | 86.4 |
| Geo-R1 | 0.0 | 1.1 | 17.6 | 59.7 | 86.9 |
| Qwen-2.5-VL-7B | 0.0 | 1.9 | 10.5 | 55.2 | 88.9 |

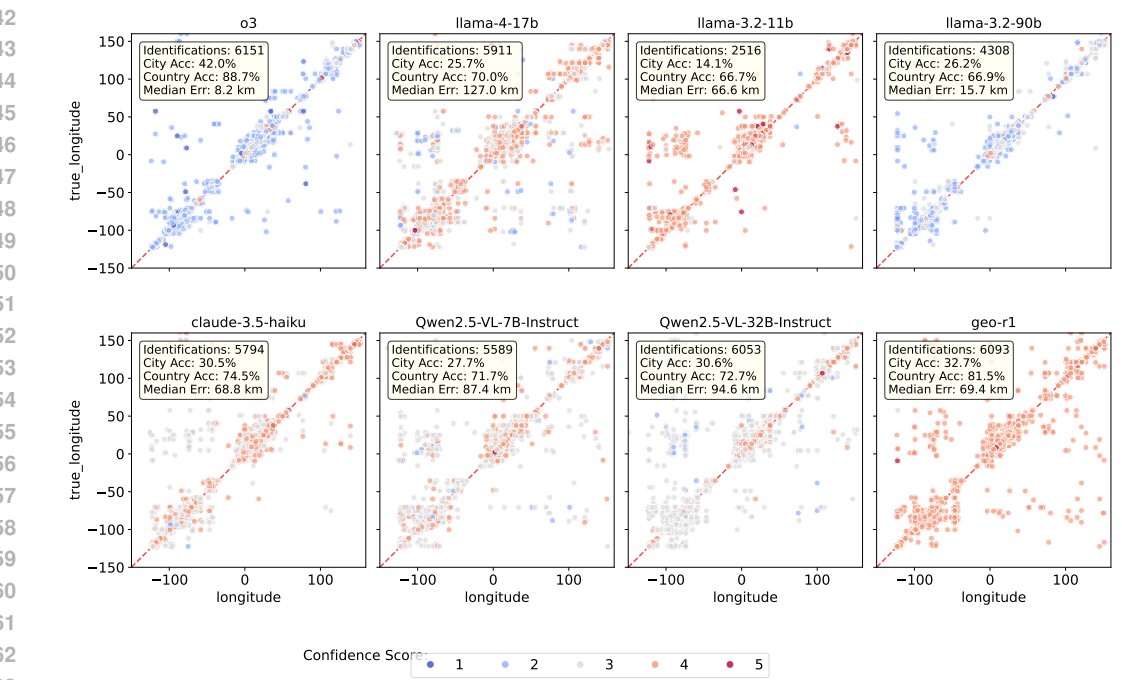

Figure 20: Longitutde analysis of IMAGEO-GSS

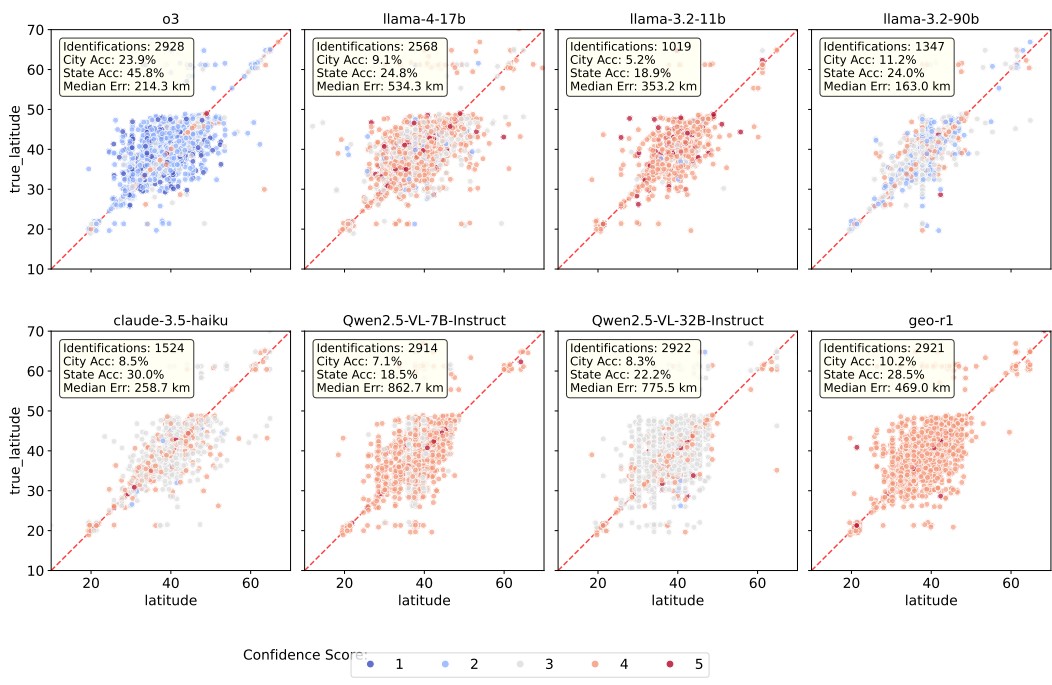

Figure 21: Latitude analysis of IMAGEO-UPC

## G  GENERAL VLM TASKS

For the general VLM benchmarks, we evaluated Geo-R1, the base model Qwen2.5-VL-7B-Instruct, as well as Geo-R1-Zero and Geo-SFT, to demonstrate this post-training process's ability to preserve the base model's original capabilities.

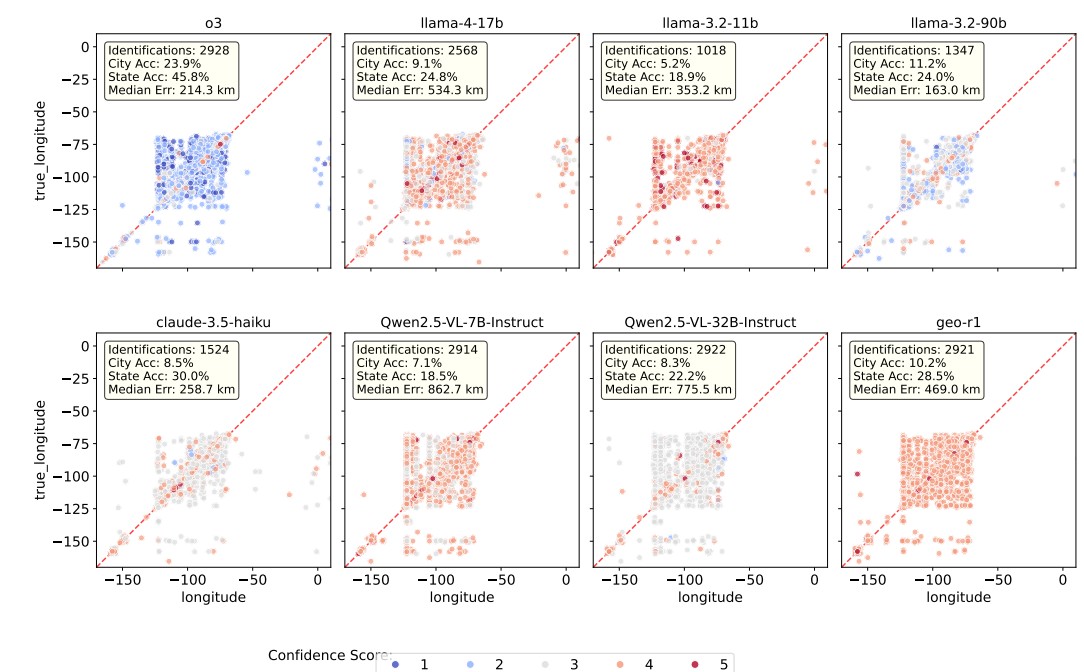

Figure 22: Longitutde analysis of IMAGEO-UPC

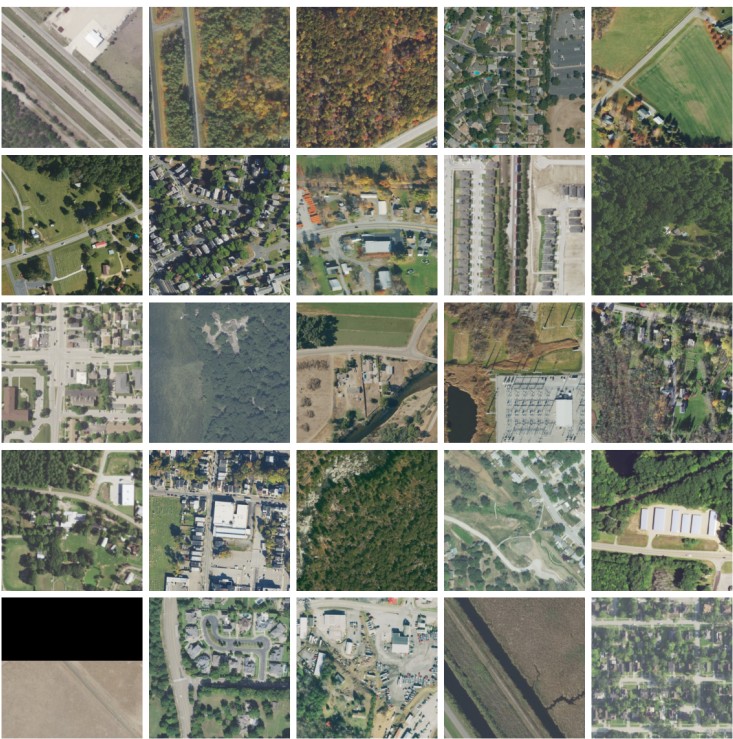

Figure 23: Random subset of NAIP images used for aerial image geolocation benchmarking, derived from Ge et al. (2025).

## G.1 MEGA-BENCH

MEGA-Bench is a large-scale multimodal benchmark comprising 8185 manually-annotated examples from 505 tasks. The dataset is designed to cover diverse real-world VLM capabilities across

varied input types (images, documents, videos, UI, infographics, etc.) and output formats (text, numbers, LaTeX, code, JSON, structured plans). Instead of relying completely on multiple-choice, it supports rich answer types evaluated with over 45 tailored metrics, combining rule-based checks with LLM-as-judge scoring for open-ended responses.

We evaluted on both the 'core' and 'open' problem sets. For all models we used 512 token max completion length. For the LLM judge, we used GPT-4o, api-version *2025-01-01-preview*. We split the evaluation using the 10 high-level tasks in the benchmark. See Table 14.

Table 14: Test Results on Mega-Bench.

| Category | Qwen2.5-VL-7B | Geo-SFT | Geo-R1-Zero | Geo-R1 |
|---|---|---|---|---|
| Commonsense and Social Reasoning | 0.46810 | 0.46181 | 0.49149 | 0.45442 |
| Domain-Specific Knowledge and Skills | 0.36163 | 0.33589 | 0.35176 | 0.34114 |
| Ethical and Safety Reasoning | 0.59229 | 0.58481 | 0.57834 | 0.56829 |
| Language Understanding and Generation | 0.43942 | 0.38218 | 0.44582 | 0.37890 |
| Mathematical and Logical Reasoning | 0.31291 | 0.23013 | 0.31707 | 0.28354 |
| Object Recognition and Classification | 0.37876 | 0.31247 | 0.40089 | 0.34824 |
| Planning and Decision Making | 0.08823 | 0.08763 | 0.15305 | 0.09632 |
| Scene and Event Understanding | 0.39181 | 0.34985 | 0.43067 | 0.38939 |
| Spatial and Temporal Reasoning | 0.24667 | 0.20043 | 0.27925 | 0.26250 |
| Text Recognition (OCR) | 0.46050 | 0.40462 | 0.44696 | 0.36919 |

## G.2 MMMU

The Massive Multi-discipline Multimodal Understanding and Reasoning Benchmark for Expert AGI (MMMU) is a large-scale benchmark of 11.5K multimodal, college-level questions spanning six disciplines, 30 subjects, and 183 subfields, using 30 image types such as diagrams, medical scans, chemical structures, sheet music, and comics. The benchmark emphasizes both breadth (coverage across many domains) and depth (expert-level reasoning difficulty). Questions, mostly multiple-choice with some open-ended, require models to integrate visual perception, domain-specific knowledge, and deliberate reasoning. We evaluate our models on the MMMU-Val and MMMU-Test sets, with 512 token max completion length. See Table 15.

Table 15: Model accuracy on MMMU Yue et al. (2024) Dev. and Validation splits.

| Model | MMMU Dev. | MMMU Val. |
|---|---|---|
| Qwen2.5-VL-7B-Instruct | 58.0 | 54.2 |
| Geo-SFT | 56.7 | 53.4 |
| Geo-R1-Zero | 50.7 | 54.3 |
| Geo-R1 | 54.0 | 51.2 |

Table 16: Model performance on GPQA benchmark results ('extended' dataset).

| Model | Accuracy (%) | Refusal Rate (%) |
|---|---|---|
| Geo-SFT | 31.1 | 1.1 |
| Geo-R1-Zero | 33.0 | 1.5 |
| Geo-R1 | 33.7 | 0.0 |
| Qwen-2.5-VL-7B | 34.2 | 3.3 |

## G.3 GPQA

GPQA is a graduate-level, expert-curated benchmark of multiple-choice questions in physics, chemistry, and biology, designed to be objective and difficult to solve via basic internet search. The dataset is compact but rigorous. Authored and validated by PhD experts, the dataset highlights challenges that lie beyond the reach of non-experts, who achieve 30–34% accuracy even with internet access, compared to experts' 72–81%. We evaluate our post-training checkpoints, editing only lightly

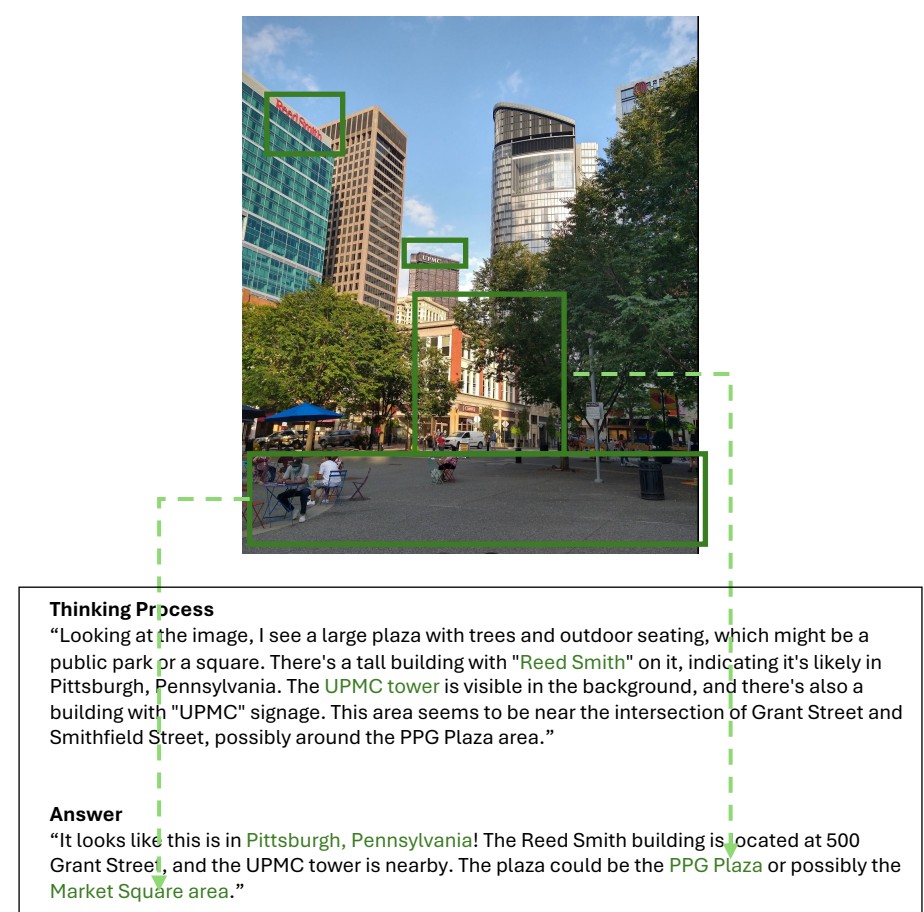

**Thinking Process**

"Looking at the image, I see a large plaza with trees and outdoor seating, which might be a public park or a square. There's a tall building with "Reed Smith" on it, indicating it's likely in Pittsburgh, Pennsylvania. The UPMC tower is visible in the background, and there's also a building with "UPMC" signage. This area seems to be near the intersection of Grant Street and Smithfield Street, possibly around the PPG Plaza area."

**Answer**

"It looks like this is in Pittsburgh, Pennsylvania! The Reed Smith building is located at 500 Grant Street, and the UPMC tower is nearby. The plaza could be the PPG Plaza or possibly the Market Square area."

Figure 24: Case study: The `Geo-R1` model can find the visual cues in the image, and link the visual cue to its hidden geospatial knowledge.

the authors evaluation code. We use thed GPQA-Extended dataset, which has 546 questions. See Table 16. For all models we used 1000 token max completion length.

## H CASE STUDY

We include some case study including a failure case to demo our `Geo-R1` model. All the model using the prompt: "Find the location of the given image." See the details in Fig. 24, 25, and 26.

## I THE USE OF LARGE LANGUAGE MODELS

Large Language Models (LLMs), specifically ChatGPT, were used as an auxiliary tool in the preparation of this paper. The assistance was limited to polishing writing from a grammatical perspective. No LLMs were used for data generation, experimental results, or research ideation. The authors take full responsibility for all contents of the paper.

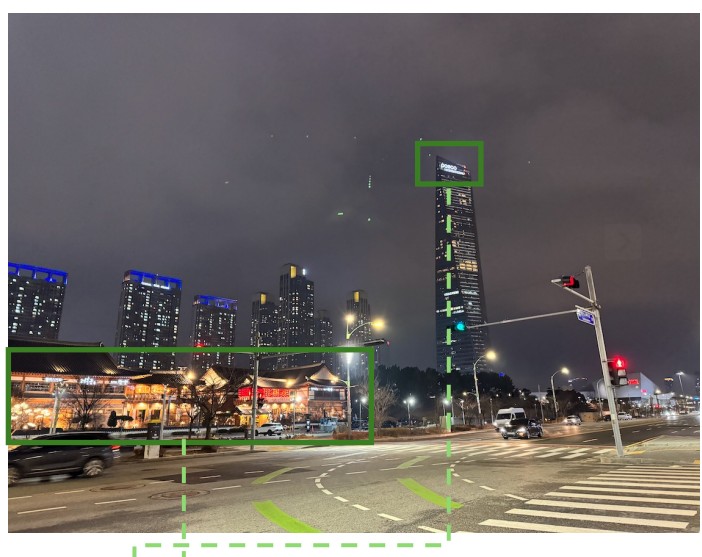

**Thinking Process**

"Alright, I'm analyzing this image. It seems to be a night scene of a city intersection with a tall tower that has "POSCO" written on it. There's also a traditional Korean building nearby, which might be a restaurant or temple. The POSCO tower is located in Songdo International Business District, Incheon, South Korea. I'm recalling that there's a traditional Korean building called "Songdo Hanok Village" near the POSCO tower. I'll check my memory for more details."

**Answer**

"It looks like the POSCO tower is in Songdo, Incheon, South Korea. The traditional Korean building could be "Songdo Hanok Village," which is close to the POSCO tower. So, I think the location is likely Songdo International Business District, Incheon."

Figure 25: Case study: The `Geo-R1` model can find the visual cues in the image, even in dark environment, where most geolocation models fail. It links the visual cue to its hidden geospatial knowledge.

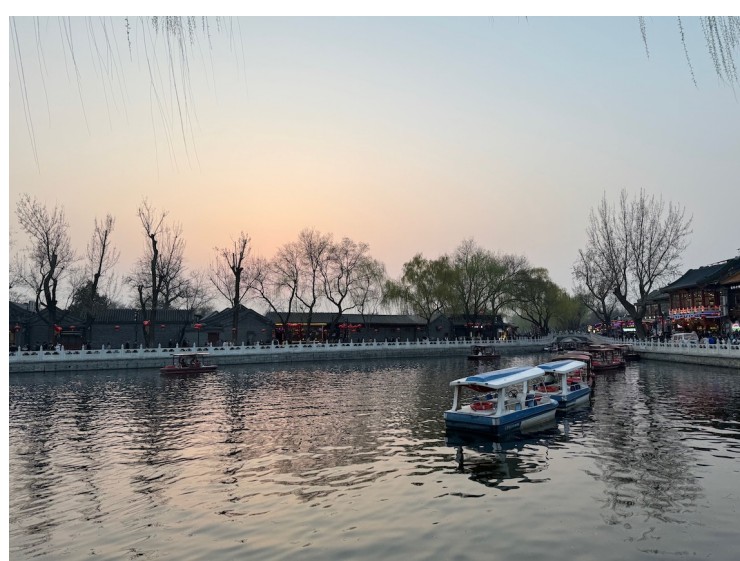

**Thinking Process**

"Looking at the image, I see a canal with boats, possibly rowboats or small motorboats, and a white railing along the bank. There's a cluster of buildings that seem to be traditional Chinese architecture, possibly with red lanterns. The trees appear to be willows, which are common in Beijing. The setting seems to be a tourist area, possibly near the Houhai lakes or the Qianmen district. I'm considering if this could be the area around the Qianmen Bridge or perhaps the Liulichang Street area."

**Over Thinking!**

**Answer**

"It looks like it could be near the Houhai lakes in Beijing, China. The canal, boats, and traditional Chinese buildings suggest it might be the area around Qianmen or Liulichang. The white railing and red lanterns add to the authentic feel!"

Figure 26: Failure case study: The `Geo-R1` model may overthink, especially when multiple locations share highly similar geographical features. In this case, it had already provided the correct answer for "Houhai Park, Beijing." However, it overthought and directed the answer to two locations in Beijing that were not precisely the one depicted in the photo..

