# OpenReview forum: "Geo-R1: Unlocking VLM Geospatial Reasoning with Cross-View Reinforcement Learning"
_ICLR.cc/2026/Conference — Submitted to ICLR 2026_

### Official Review · Reviewer_wQAM · 2025-10-25

**Soundness:** 3
**Presentation:** 3
**Contribution:** 2
**Rating:** 6
**Confidence:** 3

**Summary:**

This paper proposes Geo-R1, a post-training framework designed to equip general-purpose vision-language models with geospatial reasoning capabilities. The authors designed a two-stage training pipeline: first, infusing a geographic reasoning paradigm through scaffolding-style supervised fine-tuning, followed by reinforcement learning using verifiable rewards based on a cross-view image matching task to optimize the accuracy and structure of the reasoning. Experiments show that Geo-R1 not only achieves over 60% performance improvement on in-distribution cross-view tasks but also excels on multiple zero-shot geographic reasoning benchmarks, while largely maintaining its general multimodal capabilities without degradation.

**Strengths:**

1.	The paper accurately identifies the core challenges faced by VLMs in the geospatial reasoning domain. The proposed two-stage training framework, "SCAFFOLDING WITH SFT + RLVR," is well-designed. Experiments also demonstrate the framework's effectiveness.

2.	The evaluation system is very comprehensive, covering in-distribution performance, zero-shot out-of-distribution generalization, capability retention, and detailed ablation studies.

3.	The reward function is reasonably designed. The presentation and analysis of phenomena in the training dynamics, such as the "Aha Moment" and "dual ascent," validate the training's effectiveness.

**Weaknesses:**

1.	The core techniques of this work are a direct adaptation of paradigms from pioneering works like DeepSeek-R1, constituting combinatorial innovation. No innovation in algorithms or training methods.
2.	As mentioned in the introduction, geospatial reasoning encompasses various tasks. However, the main experiments lack a fine-grained breakdown of performance across different sub-tasks. Is performance consistent across different tasks?
3.	The article can add some case studies, especially failure case studies.

**Questions:**

1.	The core techniques of this work are a direct adaptation of paradigms from pioneering works like DeepSeek-R1, constituting combinatorial innovation. No innovation in algorithms or training methods.
2.	As mentioned in the introduction, geospatial reasoning encompasses various tasks. However, the main experiments lack a fine-grained breakdown of performance across different sub-tasks. Is performance consistent across different tasks?
3.	The article can add some case studies, especially failure case studies.

---

> ### Author Response · Authors · 2025-11-23
>
> We sincerely appreciate the constructive comments from reviewer wQAM and the time spent on reviewing this paper. We address the questions and clarify the issues accordingly as described below.
>
>
>
> >**[Weakness 1]**: The core techniques of this work are a direct adaptation of paradigms from pioneering works like DeepSeek-R1, constituting combinatorial innovation. No innovation in algorithms or training methods.
>
>
>
> **[Response to Weakness 1]**:  Although our work leverages existing SFT and GRPO-based RL techniques, these are not the core contribution of the paper. The novelty lies in how we **apply SFT and RL to proxy tasks to induce transferable reasoning patterns**, enabling the model to improve performance on downstream geospatial tasks without ever training on them. The proxy tasks and training data are entirely disjoint from the downstream evaluation, which means the model achieves not only **zero-shot data adaptation**, but also **zero-shot task adaptation.**
>
>
>
> Our key insight is that effective geospatial adaptation does not require direct fine-tuning on the target task; instead, we show that the model can generalize to downstream domains through the **construction of domain-level reasoning frameworks**. This is fundamentally different from prior RL works—for example, in mathematical reasoning—where models may be zero-shot on a dataset (e.g., training on GSM8K and testing on MATH-500) but are not zero-shot on the task itself.
>
>
>
> While the underlying algorithms and reward designs follow earlier RL paradigms, our **training philosophy** is intentionally different. The SFT stage is not meant for cold-starting or fitting sparse signals, but for instilling a **thinking paradigm** that guides subsequent reasoning, as reflected in the comparison between Geo-SFT and the original Qwen model in Table 1. Similarly, our RL stage does not train the model on the downstream geolocation task directly; instead, it supervises and shapes the model’s reasoning behavior via a proxy objective. We therefore view our approach as a novel methodology for **domain-level adaptation through reasoning construction**, rather than task-specific fine-tuning.
>
>
>
> >**[Weakness 2]**: As mentioned in the introduction, geospatial reasoning encompasses various tasks. However, the main experiments lack a fine-grained breakdown of performance across different sub-tasks. Is performance consistent across different tasks?
>
>
>
> **[Response to Weakness 2]**: We thank the reviewer for emphasizing the importance of fine-grained analysis. We respectfully clarify that a detailed breakdown of performance across 13 distinct geospatial reasoning sub-tasks (spanning Geographical, Environmental, Sociocultural, and Geolocation categories) is visually presented in Figure 1 of the main paper and numerically detailed in Table 10 of Appendix D. We include the detailed description of these sub-tasks in Table 9. We discussed the results in Section 6.3.1 and Appendix D. Regarding performance consistency, our results demonstrate that Geo-R1 achieves consistent and significant improvements over the baseline across all sub-tasks, as shown in Table 10.
>
>
>
> >**[Weakness 3]**: The article can add some case studies, especially failure case studies.
>
>
>
> **[Response to Weakness 3]**: We appreciate the reviewer’s suggestion to include qualitative case studies. In the revised manuscript, we have incorporated three representative case analyses as shown in Fig. 24, 25 and 26, with the final one explicitly focusing on a failure case to illustrate the model’s current limitations. These examples highlight both the strengths and failure modes of Geo-R1 and provide additional interpretability for the model’s reasoning behavior.

---

### Official Review · Reviewer_xBX8 · 2025-10-31

**Soundness:** 3
**Presentation:** 3
**Contribution:** 3
**Rating:** 6
**Confidence:** 4

**Summary:**

Geo-R1 proposes a reasoning-centric post-training framework designed to enhance the capabilities of Vision-Language Models in geospatial reasoning tasks. The framework consists of two stages, the scaffolding stage utilizes supervised fine-tuning on synthetic chain-of-thought data to instill a "geospatial thinking paradigm" in the model. and the elevating stage employs GRPO-based reinforcement learning using a cross-view image pairing task as a verifiable, weakly-supervised reward signal to refine the model's reasoning quality and generalization.The method achieves state-of-the-art performance on multiple geospatial reasoning benchmarks while effectively avoiding catastrophic forgetting and preserving the model's capabilities on general-purpose tasks.

**Strengths:**

1.The paper is well written and easy to follow.
2.The cross-view pairing task is both challenging and easily verifiable, making it highly suitable for weakly-supervised RL.
3. Extensive evaluation covering in-distribution, out-of-distribution generalization, and preservation of primitive abilities show the strength of the method.

**Weaknesses:**

A significant limitation is that the use of GRPO to enhance MLLM reasoning is no longer novel. The paper does not introduce substantive innovations in the core algorithm or reward mechanism, resulting in a lack of clear technical contribution to the methodology itself.but consider the written quality and solid experiments, I incline to give a positive rate.

**Questions:**

See weakness

---

> ### Author Response · Authors · 2025-11-23
>
> We sincerely appreciate the constructive comments from reviewer xBX8 and the time spent on reviewing this paper. We address the questions and clarify the issues accordingly as described below.
>
> >**[Weakness 1]**: A significant limitation is that the use of GRPO to enhance MLLM reasoning is no longer novel. The paper does not introduce substantive innovations in the core algorithm or reward mechanism, resulting in a lack of clear technical contribution to the methodology itself.
>
> **[Response to Weakness 1]**: Although our work leverages existing SFT and GRPO-based RL techniques, these are not the core contribution of the paper. The novelty lies in how we **apply SFT and RL to proxy tasks to induce transferable reasoning patterns**, enabling the model to improve performance on downstream geospatial tasks without ever training on them. The proxy tasks and training data are entirely disjoint from the downstream evaluation, which means the model achieves not only **zero-shot data adaptation**, but also **zero-shot task adaptation.**
>
> Our key insight is that effective geospatial adaptation does not require direct fine-tuning on the target task; instead, we show that the model can generalize to downstream domains through the **construction of domain-level reasoning frameworks**. This is fundamentally different from prior RL works—for example in mathematical reasoning—where models may be zero-shot on a dataset (e.g., training on GSM8K and testing on MATH-500) but are not zero-shot on the task itself.
>
> While the underlying algorithms and reward designs follow earlier RL paradigms, our **training philosophy** is intentionally different. The SFT stage is not meant for cold-starting or fitting sparse signals, but for instilling a **thinking paradigm** that guides subsequent reasoning, as reflected in the comparison between Geo-SFT and the original Qwen model in Table 1. Similarly, our RL stage does not train the model on the downstream geolocation task directly; instead, it supervises and shapes the model’s reasoning behavior via a proxy objective. We therefore view our approach as a novel methodology for **domain-level adaptation through reasoning construction**, rather than task-specific fine-tuning.

---

> > ### Comment · Reviewer_xBX8 · 2025-11-27
> >
> > Thank you for your reply, I have no other questions, I will keep my positive rate.

---

### Official Review · Reviewer_hY66 · 2025-11-01

**Soundness:** 2
**Presentation:** 2
**Contribution:** 2
**Rating:** 2
**Confidence:** 4

**Summary:**

The paper proposes Geo-R1, a reasoning-first reinforcement learning framework for geospatial understanding tasks. It uses synthetic chain-of-thought supervision and cross-view alignment as weak rewards to fine-tune a base VLM (Qwen2.5-VL-7B). Experiments on several benchmarks show improvements over open-source general-purpose VLMs.

**Strengths:**

Adapts an RL-based approach for geospatial reasoning.

Builds a large and diverse synthetic dataset and applies CoT reasoning signals effectively.

Achieves measurable gains on public benchmarks like GeoChain and IMAGEO-Bench.

**Weaknesses:**

Baseline coverage is incomplete: The paper only compares against general VLMs (Qwen, LLaMA, Claude) but omits numerous strong task-specific geolocalization and remote-sensing models (e.g., GeoCLIP). These are directly relevant to the core tasks evaluated.

All experiments are conducted solely on Qwen2.5-VL-7B, no results are provided for other architectures. This makes it difficult to assess the generalizability or stability of the proposed RL pipeline.

**Questions:**

Have the authors attempted to apply the same RL pipeline to other backbones (e.g., LLaVA-Next, InternVL, Phi-3-V)?

Why were task-specific geolocalization baselines not included? Many of them have public weights and could provide stronger comparisons.

An anonymous GitHub repository is provided, but it is currently empty. The authors should not claim code availability or include the link unless the source code is indeed provided at submission time.

---

> ### Author Response · Authors · 2025-11-23
>
> We sincerely appreciate the constructive comments from reviewer hY66 and the time spent on reviewing this paper. We address the questions and clarify the issues accordingly as described below.
>
> >**[Weakness 1]**: Baseline coverage is incomplete: The paper only compares against general VLMs (Qwen, LLaMA, Claude) but omits numerous strong task-specific geolocalization and remote-sensing models (e.g., GeoCLIP). These are directly relevant to the core tasks evaluated.
>
> **[Response to Weakness 1]**: We thank the reviewer for pointing this out. We agree that geolocalization-specific baselines are important complements to general-purpose VLM comparisons. In the revised version, we have expanded our baseline coverage accordingly.
>
> We have added experiments evaluating Geo-R1 against GeoCLIP, a state-of-the-art location-image alignment model that is directly relevant to global geo-localization, considering its weights are publicly available. As shown in Table 2 and in Section 6.3 of the updated paper, Geo-R1 significantly outperforms GeoCLIP across all metrics on the IMAGEO-GSS benchmark, including city-level and country-level accuracy as well as distance error. This demonstrates that our model's geolocation capabilities in **complex environmental scenarios** far surpass those of traditional models like GeoCLIP.
>
> To further strengthen the comparison, we also evaluate on the IM2GPS3K benchmark, a canonical testbed for task-specific geolocation models. As shown in Section 6.3 and Table 3, we compare Geo-R1 against a series of task-specific geolocation models that represent the core set of classical and modern non-LLM geolocalization models. As shown in Table 3, our model—in zero-shot mode—achieves competitive or superior performance across multiple localization thresholds (1km–2500km), and when enabling thinking mode, Geo-R1 achieves the strongest results. This demonstrates the superiority of our model over non-LLM approaches.
>
> >**[Weakness 2]**: All experiments are conducted solely on Qwen2.5-VL-7B, no results are provided for other architectures. This makes it difficult to assess the generalizability or stability of the proposed RL pipeline.
>
> **[Response to Weakness 2]**: We thank the reviewer for raising this concern. While our initial experiments focused on Qwen2.5-VL-7B for clarity and efficiency, the revised manuscript now includes results on a different and stronger architecture, InternVL3-8B, demonstrating that our RL pipeline is not tied to a specific backbone. As shown in the newly added Table 4 and Section 6.6, applying our GRPO-based post-training to InternVL3-8B yields substantial performance improvements (+64\% in distribution cross-view pairing accuracy, +16.8\% out-of-distribution city accuracy and large reductions in distance error). This confirms that our RL pipeline is not tied to the Qwen architecture and provides consistent benefits across different model families. Qwen2.5-VL-7B was chosen as the primary model due to its open licensing and efficient handling of multi-image inputs, but the extended results now clearly validate that our method generalizes beyond a single model and works robustly on other high-performing VLM architectures.
>
> >**[Question 1]**: Have the authors attempted to apply the same RL pipeline to other backbones (e.g., LLaVA-Next, InternVL, Phi-3-V)?
>
> **[Response to Question 1]**: See Response to Weakness 2. In newly added Table 4 and Section 6.6, we now include results on InternVL-3 in our paper. We are actively applying our pipeline to other backbones like Phi-4-multimodal and LLaVA series. We will include the results in the paper once the training is finished.
>
> >**[Question 2]**: Why were task-specific geolocalization baselines not included? Many of them have public weights and could provide stronger comparisons.
>
> **[Response to Question 2]**: See Response to Weakness 1.
>
> >**[Question 3]**: An anonymous GitHub repository is provided, but it is currently empty. The authors should not claim code availability or include the link unless the source code is indeed provided at submission time.
>
> **[Response to Question 3]**: Thank you for pointing this out. **The anonymous link has now been fixed and is fully accessible.**
> It appears that the initial upload did not synchronize correctly to the anonymous repository, which caused the temporary access issue.
> We apologize for the inconvenience. We reiterate that the **code** for the model pipeline and the model **weights** include Geo-R1, Geo-SFT and Geo-R1-Zero will be fully open-source.

---

### Official Review · Reviewer_i8Rs · 2025-11-01

**Soundness:** 3
**Presentation:** 3
**Contribution:** 3
**Rating:** 4
**Confidence:** 4

**Summary:**

This paper adopts an SFT stage as a cold-start to warm up the MLLM model to handle geospatial CoT capacities by curating a synthetic dataset. Then, the authors design a cross-view selection reward pipeline to perform the GeoSpatial RL post-training based on the post-training dataset. Meanwhile, this paper conducts SFT, then RL and direct RL manner to train MLLM, like Qwen2.5VL, to evaluate the geospatial generalization abilities of the model. Through careful reward function adaptations, this paper obtains competitive cross-view geospatial evaluation performances.

**Strengths:**

1. The application of adopting MLLM like Qwen2.5VL to geospatial tasks sounds like an interesting application.

2. The post-training pipeline based on MLLM seems consistent with most RL-needed tasks.

3. The final zero-shot generalization in terms of in-distribution and out-of-distribution performances is competitive and promising.

**Weaknesses:**

1. Though the final geospatial performances are competitive, the overall training receipts or designs are not that novel, which sounds like an application from the RL GRPO to these new downstream tasks.

2. I am confused about the reward score assignments for the cross-view selection objective, while the correct one gets a 1.0 score, and all the others get -0.8. Does the author demonstrate well about this choice? What if we set 0 for the intermediate options?

3. As from the main paper, the visual appearances among the geo image and the satellite images, does the paper also make use of other metadata to help the model learning, since the large differences are not that reasonable to help the model learning towards mapping between the correct geospatial images and the satellite candidates?

4. Regarding Figure 8, the figure here looks quite blurry. I suggest that the authors consider inserting better presentations.

**Questions:**

The experimental results are limited, and the comparisons with other MLLMs (either open or closed sources) are very limited in this paper. What if compared with Intern3VL, GPT-4O, and O3? And also, the ablative studies seem very few to understand the effective post-training results in this paper, for example, some ablations about the cross-view between spatial image and satellite images, by changing the satellite images with other mixed spatial images?

---

> ### Author Response · Authors · 2025-11-23
>
> We sincerely appreciate the constructive comments from reviewer i8Rs and the time spent on reviewing this paper. We address the questions and clarify the issues accordingly as described below.
>
> >**[Weakness 1]**: Though the final geospatial performances are competitive, the overall training receipts or designs are not that novel, which sounds like an application from the RL GRPO to these new downstream tasks.
>
> **[Response to Weakness 1]**:  Although our work leverages existing SFT and GRPO-based RL techniques, these are not the core contribution of the paper. The novelty lies in how we **apply SFT and RL to proxy tasks to induce transferable reasoning patterns**, enabling the model to improve performance on downstream geospatial tasks without ever training on them. The proxy tasks and training data are entirely disjoint from the downstream evaluation, which means the model achieves not only **zero-shot data adaptation**, but also **zero-shot task adaptation.**
>
> Our key insight is that effective geospatial adaptation does not require direct fine-tuning on the target task; instead, we show that the model can generalize to downstream domains through the **construction of domain-level reasoning frameworks**. This is fundamentally different from prior RL works—for example in mathematical reasoning—where models may be zero-shot on a dataset (e.g., training on GSM8K and testing on MATH-500) but are not zero-shot on the task itself.
>
> While the underlying algorithms and reward designs follow earlier RL paradigms, our **training philosophy** is intentionally different. The SFT stage is not meant for cold-starting or fitting sparse signals, but for instilling a **thinking paradigm** that guides subsequent reasoning, as reflected in the comparison between Geo-SFT and the original Qwen model in Table 1. Similarly, our RL stage does not train the model on the downstream geolocation task directly; instead, it supervises and shapes the model’s reasoning behavior via a proxy objective. We therefore view our approach as a novel methodology for **domain-level adaptation through reasoning construction**, rather than task-specific fine-tuning.
>
>
> >**[Weakness 2]**: I am confused about the reward score assignments for the cross-view selection objective, while the correct one gets a 1.0 score, and all the others get -0.8. Does the author demonstrate well about this choice? What if we set 0 for the intermediate options?
>
>
> **[Response to Weakness 2]**: Our accuracy reward is designed as follows: If the model provides an answer that is correct, it receives a positive reward of 1 point; if it provides an answer that is incorrect, it incurs a penalty of -0.8 points; if the model fails to follow the instruction and provide any answer at all, it incurs a penalty of -1 point. The distinction between -0.8 for incorrect answers and -1 for no response stems from our consideration to prevent reward hacking. If both were assigned -1, the model might opt to refuse answering altogether when faced with prolonged inability to provide responses. This would cause the RL process to collapse. So, we give a little reward when the model starts giving answers. On the other hand, incorrect is incorrect, so we think -0.8 is a good choice between [-1,1]. This choice is intuitive, yet reasonable.
>
> The reward scale itself is not unique, as GRPO employs relative advantage. $\hat{A}_{i,t} = \tilde{r}_i = \frac{r_i - \mathrm{mean}(r)}{\mathrm{std}(r)}
> $. The calculation method for advantage in GRPO effectively normalizes the reward. Due to GRPO’s reward normalization, our method is not sensitive to the exact raw values as long as the correct option receives a higher reward than the incorrect ones. This means using {-1, -0.8, 1} versus {0, 0.1, 1} or {-10,-8,10} yields no difference in advantage. We ultimately selected the {-1, -0.8, 1} after considering its alignment with other rewards (format {0,1}, length [-1,0.5], and repetition [-1,0]).
>
>
> As for why we uniformly applied -0.8 to incorrect answers instead of using a differentiated reward based on the degree of error, for instance, giving an intermediate reward like 0 or -0.2 for a closer but incorrect option, our consideration is as follows: This type of interpolation is difficult to define because: (1) The proxy task we selected is inherently challenging, as both the chosen incorrect answers and the correct answer are very close, making it hard to judge which incorrect option is closer. (2) This indirect reward for judging the direction of action may lead to reward hacking, causing instability in the training process.

---

> > ### Author Response · Authors · 2025-11-23
> >
> > >**[Weakness 3]**: As from the main paper, the visual appearances among the geo image and the satellite images, does the paper also make use of other metadata to help the model learning, since the large differences are not that reasonable to help the model learning towards mapping between the correct geospatial images and the satellite candidates?
> >
> >
> > **[Response to Weakness 3]**: We thank the reviewer for raising this concern. We would like to clarify that our method does not use any metadata, geo-coordinates, EXIF information, headings, timestamps, or any auxiliary signals. All training and inference rely solely on the raw visual contents of the ground-level images and satellite candidates.
> >
> > Although geo and satellite views exhibit large viewpoint differences, prior literature (e.g., DSM[1], VIGOR[2]) consistently shows that stable visual cues, such as road topology, building layout, coastline shape, vegetation distribution, and urban density, provide sufficient information to learn cross-view correspondences.
> >
> > [1] Where Am I Looking At? Joint Location and Orientation Estimation by Cross-View Matching. In CVPR 2020.
> >
> > [2] Sijie Zhu, Taojiannan Yang, Chen Chen. VIGOR: Cross-View Image Geo-localization beyond One-to-one Retrieval. In CVPR 2021.
> >
> > >**[Weakness 4]**:  Regarding Figure 8, the figure here looks quite blurry. I suggest that the authors consider inserting better presentations.
> >
> > **[Response to Weakness 4]**: We thank the reviewer for pointing this out.
> > In the revised manuscript, we have re-rendered Figure 8 from the raw logged scalar values and replaced the previous image with high-resolution vector graphics (PDF format). We also increased line clarity, and improved the layout. The updated figure is now crisp and readable at any zoom level. Similarly, for Figures 9 to 18, we are expediting the creation of new vector graphics, which will be updated in the coming days.
> >
> > >**[Question 1]**: The experimental results are limited, and the comparisons with other MLLMs (either open or closed sources) are very limited in this paper. What if compared with Intern3VL, GPT-4O, and O3?
> >
> > **[Response to Question 1]**: We include our the comparison with o3 in Table 11, 12 and Fig. 5, 19, 20, 21 and 22. We appreciate the suggestion to include InternVL and GPT. We have now evaluated InternVL-3 and GPT-4o on the IMAGEO-GSS benchmark to ensure a comprehensive comparison. The results of InternVL3 and GPT-4o are now included in Table 11. The results demonstrate that our model still achieves superior performance when compared to these models, considering that Geo-R1 has only 7 billion parameters. In the paper, we also include comparisons of multiple versions of MLLMs such as Llama, Qwen, and Claude, spanning a wide range. In addition, we have introduced comparisons with multiple specialized geolocation models. As detailed in Tables 2 and 3, our model outperforms traditional task-specific geolocation models in both conventional and complex geolocation tasks.
> >
> > >**[Question 2]**: And also, the ablative studies seem very few to understand the effective post-training results in this paper, for example, some ablations about the cross-view between spatial image and satellite images, by changing the satellite images with other mixed spatial images?
> >
> > **[Response to Question 2]**: Thank you for rasing an interesting point about testing robustness by "changing satellite images with mixed spatial images." In fact, during the experimental design phase, we attempted to use mixed spatial images. However, the significant disparity in difficulty across different spatial modalities led to instability in the model's accuracy reward. This is fatal to the RL training process. We are considering a curriculum-based learning approach, which we reserve for future work.
> >
> > We believe our current training setup already addresses the core of this concern through "Hard Negative" Mining, which acts as a rigorous ablation of the model's discriminative ability: In our RL stage (Section 5.1), we do not just use random "mixed spatial images" (which might be easily distinguishable by modality alone, e.g., street view vs. aerial). Instead, we specifically use confusers from the same city. This forces the model to ignore modality shifts and focus on fine-grained semantic correspondence (e.g., specific building shapes, road layouts) rather than just learning to classify "this is a satellite image." As shown in Table 1 (Ablation of Training Stages), the base model starts at random guess accuracy (~19\%). SFT improves this marginally to 23\%, but the RL stage (with these hard negatives) boosts accuracy to 82.4\%. This confirms that the specific design of our cross-view pairing reward is the key driver of performance.

---

### Author Response · Authors · 2025-12-03

Dear Area Chair and Reviewers,

We sincerely thank you for the time spent reviewing our work and for your constructive comments.

During the rebuttal period, we have actively engaged with all reviewers to address concerns regarding novelty, baselines, and generalization. All reviewers' questions have been answered. We have extensively revised the manuscript (updates marked in blue). Below is a summary of the key improvements:

1. Clarification on Novelty: Reasoning-First Post-Training (Addressing i8Rs, xBX8, wQAM) We clarified that our contribution is not merely applying SFT and GRPO, but rather the training philosophy of using proxy tasks to induce transferable reasoning patterns.

Zero-Shot Task Adaptation: Unlike prior RL works (e.g., in Math) that fit specific datasets, Geo-R1 achieves zero-shot task adaptation. The model is never trained on the downstream geolocation tasks but learns to solve them by constructing a domain-level reasoning framework.

Methodology: We emphasize that the SFT stage instills a "thinking paradigm" rather than fitting sparse signals, and the RL stage supervises reasoning behavior via a proxy objective, not the ground truth of the target task.

2. Comprehensive New Baselines (Addressing i8Rs, hY66) To validate SOTA performance, we added comparisons against significantly stronger models (see Tables 2, 3, 11, 12):

Closed-Source LLMs: As suggested by Reviewer i8Rs, we added GPT-4o and o3. Geo-R1 (7B) remains competitive or superior to these much larger models in specialized geospatial tasks.

Stronger Open-Source VLMs: We added comparisons with InternVL-3.

Task-Specific Models: Addressing Reviewer hY66's concern, we added GeoCLIP (SOTA location-image alignment) and benchmarks on IM2GPS3K. Geo-R1 significantly outperforms GeoCLIP in city/country-level accuracy and distance error, proving its superiority over traditional non-LLM approaches.

3. Generalization Across Architectures (Addressing hY66) To prove our pipeline is not architecture-specific (a concern raised by Reviewer hY66), we applied our GRPO-based post-training to InternVL3-8B (added in Table 4, Section 6.6).

Result: The method yielded substantial gains (+64% in-distribution, +16.8% OOD accuracy), confirming that our "Reasoning-First" pipeline is robust and transferable to other high-performing VLM backbones.

4. Enhanced Analysis & Fixes (Addressing i8Rs, wQAM, hY66)

Reward Design (i8Rs): We provided a detailed justification for our {-1, -0.8, 1} reward scheme, explaining how it prevents reward hacking (refusal to answer) while maintaining the relative advantage required by GRPO.

Qualitative Analysis (wQAM): We added case studies (including failure cases) in Figs 24-26 to improve interpretability.

Visuals (i8Rs): We re-rendered Figure 8 and others as high-resolution vector graphics.

Code (hY66): The anonymous GitHub link has been fixed and is fully accessible.

We believe these updates effectively address the concerns raised. We are confident that Geo-R1 represents a significant step forward in enabling VLMs to "reason" about the physical world without expensive supervision.

Best regards,

The Authors

---

### Meta-Review · Area_Chair_n1bd · 2025-12-21

**Summary:**

Geo-R1 applies a two-stage SFT + GRPO RL pipeline to induce geospatial reasoning in VLMs via synthetic CoT and cross-view pairing rewards. While empirically strong, reviewers judged the approach as largely an application of existing reasoning-RL paradigms.

Pros
* Strong in-distribution and OOD results; effective proxy reward design.
* Improved baseline coverage and added backbone experiments in rebuttal.
* Clear writing and thorough responses.

Cons
* Insufficient methodological novelty beyond prior SFT + GRPO reasoning frameworks.
* Contribution framed as “training philosophy” was not compelling as a technical advance.
* Initial gaps in baselines/architectures reduced confidence.

Overall, well-executed with solid results, but incremental; novelty concerns led to rejection.

**Reviewer Scores:**

n/a

---

### Decision · Program_Chairs · 2026-01-26

Reject